# OUT-OF-DISTRIBUTION DETECTION WITH NEGATIVE PROMPTS

**Jun Nie**[1]  **Yonggang Zhang**[2] *  **Zhen Fang**[3]  **Tongliang Liu**[4]  **Bo Han**[2]  **Xinmei Tian**[1,5] *

[1]University of Science and Technology of China    [2]TMLR Group, Hong Kong Baptist University
[3]University of Technology Sydney    [4]Sydney AI Centre, The University of Sydney
[5]Institute of Artificial Intelligence, Hefei Comprehensive National Science Center

## ABSTRACT

Out-of-distribution (OOD) detection is indispensable for open-world machine learning models. Inspired by recent success in large pre-trained language-vision models, e.g., CLIP, advanced works have achieved impressive OOD detection results by matching the *similarity* between image features and features of learned prompts, i.e., positive prompts. However, existing works typically struggle with OOD samples having features similar to those of known classes. One straightforward approach is to introduce negative prompts to achieve a *dissimilarity* matching, which further assesses the anomaly level of image features by introducing the absence of specific features. Unfortunately, our experimental observations show that employing a prompt like "not a photo of a" or learning a shared prompt for all classes fails to capture the dissimilarity for identifying OOD samples. The failure may be attributed to the diversity of negative features, i.e., tons of features could indicate features not belonging to a known class. To this end, we propose to learn a set of negative prompts for each class. The learned positive prompt (for all classes) and negative prompts (for each class) are leveraged to measure the similarity and dissimilarity in the feature space simultaneously, enabling more accurate detection of OOD samples. Extensive experiments are conducted on diverse OOD detection benchmarks, showing the effectiveness of our proposed method.

## 1 INTRODUCTION

Deep neural networks have achieved remarkable success with large-scale labeled data available. Most deep learning methods are designed in closed-set environments, where models are trained under the in-distribution (ID) assumption that the label space at test time remains the same as that of the training samples (Huang et al., 2017). However, in reality, samples from new classes may emerge spontaneously, thereby breaking this assumption. To address this issue, out-of-distribution (OOD) detection (Bendale & Boult, 2016) is gaining more attention. In OOD detection, the model is required not only to classify ID samples precisely but also to discern OOD samples accurately.

Many representative OOD detection methods are based on the post-hoc strategy (Hendrycks & Gimpel, 2017; Liang et al., 2018; Liu et al., 2020; Sun et al., 2021; Sun & Li, 2022), which identifies OOD samples by analyzing the properties of predictions made by a well-trained model on ID samples. However, the performance of post-hoc methods heavily relies on the quality of the extracted features from the well-trained model. In general, identifiable features for ID classification do not necessarily aid in the detection of OOD samples. For example, in the ID classification tasks for cats and horses, it is clear that the cat ears and horseshoes are the identifiable semantic features for ID classification, however, these features are not always applicable for OOD detection, especially when other animals, e.g., tigers, with ears similar to cat ears are involved as OOD, see Figure 1. The gap between classification and OOD detection is that in classification, we only need to consider the difference between a cat and a horse, where we can say the object with a pair of pointed ears is a cat. But in OOD detection, we need to consider the difference between a cat and all the remaining classes, where we should say the object with a pair of pointed ears, a pudgy face, a long beard and other corresponding features is a cat, and the object without a pair of pointed ears or without a pair of

---

*Correspondence to: Yonggang Zhang (csygzhang@comp.hkbu.edu.hk); Xinmei Tian (xinmei@ustc.edu.cn)

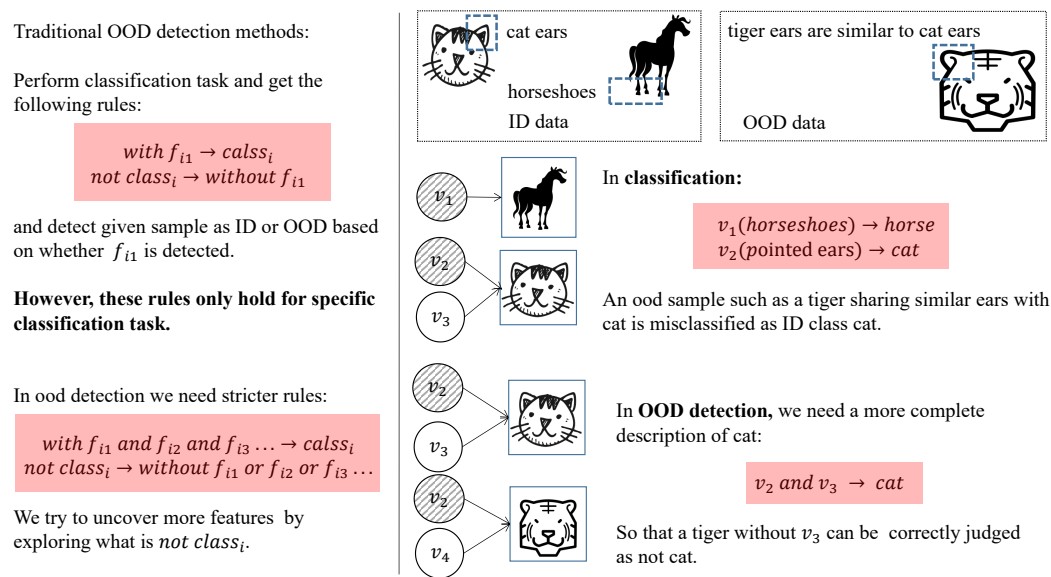

Figure 1: The features learned by ID samples for ID classification are insufficient for OOD detection. Cat ears and horseshoes are enough to distinguish between cats and horses, but they cannot distinguish between the ID class cat and the OOD class tiger, as tigers have ears similar to those of cats.

pointed ears or without long beard or other corresponding features is not a cat. Inspired by the causal understanding in (Zhang et al., 2022), we argue that simply using features learned from classification tasks under limited classes for OOD detection is not reasonable.

An effective solution involves utilizing large models like CLIP (Radford et al., 2021) trained on extensive data. Unlike models tailored for specific classification tasks on ID samples, CLIP can generate distinctive image features for all classes, both ID and OOD. This raises a pivotal question: *how can we leverage CLIP's extracted features for OOD detection?* Some previous works (Ming et al., 2022; Fort et al., 2021) propose an approach by matching the similarity between image features and text features of ID classes. Specifically, hand-crafted or learned prompts like "a photo of a [class]" are fed to the text encoder to compute text features. The cosine similarity between these text features and the image feature determines the sample's likelihood of belonging to "[class]."

One of the primary limitations of this approach lies in its exclusive reliance on the positive features of ID classes. For example, When the given image input is a tiger, and we provide "a photo of a cat" as a prompt, CLIP will assign a high score because the tiger shares similar image features, like ears, with a cat. It completely neglects the distinguishing features that differentiate a tiger from a cat. To address this challenge, we propose to construct negative prompts such as "not a photo of a " to achieve a dissimilarity matching. Our intuition is to leverage negative prompts like "not a photo of a" to identify objects absent in an image. For instance, we consider examples like "not a photo of a cat". With this prompt, our objective is to steer CLIP's attention toward features in the image unrelated to cats (i.e., negative features). It is worth noting that the description "not a photo of a cat" is an incorrect prompt for cats, but an accurate one for tigers. However, crafting effective negative prompts is not a straightforward task. Our empirical results show that simply using "not a photo of a cat" as a negative prompt leads to a higher matching score with a photo of a cat than with a photo of a tiger. This aligns with the findings of a recent study (Yüksekgönül et al., 2022).

Previous works have focused on prompt representation learning, such as CoOp (Zhou et al., 2022b) and CoCoOp (Zhou et al., 2022a), where they aim to improve image classification accuracy based on pre-trained vision-language models. In contrast to these works, our work targets learning negative prompts that inform the network of what is "not." To mitigate this challenge, in this paper, we propose a novel method named Learn to Say No (LSN), which can learn suitable negative prompts to tell the network what is "not." Unlike positive prompt learning, where the learned positive prompts are usually shared across different classes as most positive information about a class is contained in the class name, the negative information about a class can not be carried by class name alone. The negative features of a class are usually diverse. To this end, we learn a set of negative prompts for

"a photo of a [class]"

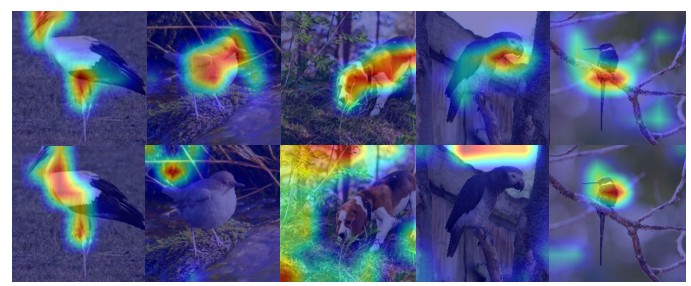

"not a photo of a [class]"

Figure 2: Qualitative comparison of the learned positive prompts and negative prompts.

each class. To further improve performance, we also learn positive prompts to replace hand-designed prompts such as "a photo of a". The simultaneous incorporation of positive and negative prompts contributes to a more accurate OOD uncertainty estimation of the pre-trained CLIP model. As shown in Figure 2, we visualize heat maps depicting the learned positive and negative prompts. These visualizations reveal distinct regions of interest for positive and negative prompts. Using both positive and negative prompts to detect samples allows focusing on more diverse features of the samples, thus achieving better OOD detection performance.

The work most similar to ours is RPL (Chen et al., 2020) and CLIPN (Wang et al., 2023a). RPL learns reciprocal points as negative representations corresponding to each known class. However, RPL constructs the information of "not belong to a class" using features learned from classification tasks on ID classes. This approach has been demonstrated to be less rational. CLIPN leverages a large-scale dataset CC3M (Sharma et al., 2018) to fine-tune the text encoder, endowing CLIP with the ability to say "no". In contrast, in this paper, we focus on learning prompts to uncover the encoded knowledge of negative features, which benefits from the powerful visual representation capabilities of CLIP and can be done in a very short time. We summarize our contributions as follows:

- We propose to use negative prompts to answer positively what is "not" to complement the feature representations, thus improving the ability of the model to handle OOD samples.

- We reveal that CLIP cannot understand the meaning of "not" very well. For this reason, we learn a set of negative prompts rather than using a hand-designed prompt.

- Extensive experiments show that our learned prompts greatly improve the OOD uncertainty estimation of CLIP, and an ablation study is conducted to understand the efficacy of LSN.

## 2 PROBLEM SETUP

Let $\mathcal{X}$ and $\mathcal{Y} = \{l_1, ..., l_C\}$ be the image space and ID label space, respectively. Note that $\mathcal{Y}$ is a set containing words, e.g., $\mathcal{Y} = \{\text{cat}, \text{dog}, \cdots, \text{horse}\}$, and $C$ is the number of ID classes. Given the ID feature random variable $X^{\text{in}} \in \mathcal{X}$ and OOD feature random variable $X^{\text{out}} \in \mathcal{X}$, we use the $\mathbb{P}_{X^{\text{in}}}$ and $\mathbb{P}_{X^{\text{out}}}$ to represent the ID marginal distribution and OOD marginal distribution, respectively.

**Image and text encoders with CLIP-like model.** Given any input image $\mathbf{x} \sim \mathbb{P}_{X^{\text{in}}}$ and any label $l_c \in \mathcal{Y}$, we extract features of $\mathbf{x}$ and $l_c$ using the CLIP-like model $\mathbf{f}$, which consists of image encoder $\boldsymbol{f}$ and text encoder $\boldsymbol{g}$. Then, the extracted features are $\boldsymbol{f}(\mathbf{x})$ and $\boldsymbol{g} \circ \boldsymbol{V}(l_c), \forall c = 1, 2, ..., C$, where $\boldsymbol{V}(\cdot)$ represents the positive prompt template for ID labels and $\circ$ means the function composition.

**OOD detection.** In the classical classification task, we assume that each image will definitely belong to a certain ID class. In OOD detection, the test samples may contain some unknown images $\mathbf{x} \sim \mathbb{P}_{X^{\text{out}}}$, and the categories of these images are beyond ID label space, i.e., $l \notin \mathcal{Y}$.

Given ID training samples $T = \{(\mathbf{x}_1, y_1), ..., (\mathbf{x}_n, y_n)\}$ drawn from $\mathbb{P}_{X^{\text{in}}}$, OOD detection aim to learn an OOD predictor $G$ using $T$, such that 1) the predictor $G$ can classify ID samples into correct ID classes and 2) the predictor $G$ can detect OOD samples as OOD.

**Score functions.** Following many representative OOD detection methods (Hendrycks & Gimpel, 2017; Liang et al., 2018; Liu et al., 2020), given a threshold $\gamma$ and a score function $S$, then $\mathbf{x}$ is detected as ID sample if and only if $S(\mathbf{x}) \geq \gamma$:

$$G_{\gamma}(\mathbf{x}) = \text{ID, if } S(\mathbf{x}) \geq \gamma; \text{ otherwise, } G_{\gamma}(\mathbf{x}) = \text{OOD.} \tag{1}$$

The performance of OOD detection depends on how to design a score function $S$ to make OOD samples obtain lower scores while ID samples have higher scores.

## 3 PROPOSED ALGORITHM

### 3.1 CAN CLIP UNDERSTAND THE MEANING OF "NOT"?

While CLIP has demonstrated impressive performance in various zero-shot and few-shot tasks, recent research (Yüksekgönül et al., 2022) reveals that state-of-the-art VLMs operate like bags-of-words, lacking relational understanding leading to errors in linking objects to their attributes, and showing a severe lack of order sensitivity. Our study further underscores CLIP's challenge in comprehending the concept of "not." As depicted in Figure 3, when presented with an image of an elephant, CLIP surprisingly ranks "not a photo of an elephant" as a more favorable match than "not a photo of a dog," contrary to reality. Additionally, in Figure 4, when neither of the two provided text inputs accurately describes the picture, CLIP tends to assign a higher probability to the input containing the word "not." From these observations, it is inferred that CLIP does not completely comprehend the meaning of "not." This may be attributed to the training method utilized by CLIP.

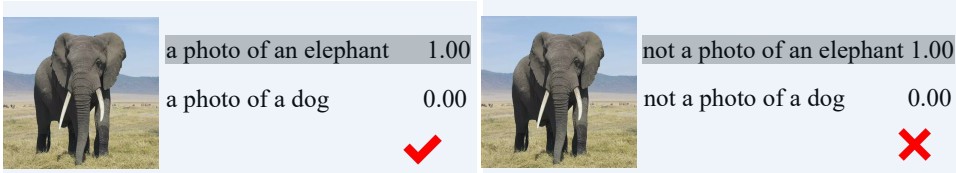

Figure 3: Each time, we feed CLIP with two text inputs and ask it to make a choice based on the given image. When the given image is an elephant and the given text inputs are "not a photo of an elephant" and "not a photo of a dog," CLIP makes the wrong choice.

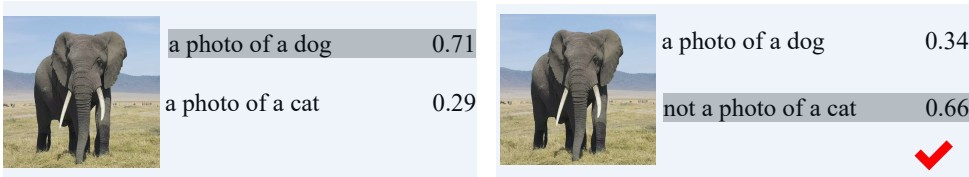

Figure 4: In the two experiments before (left) and after (right), just adding "not" to the low-scoring text input, CLIP makes a different judgment, indicating that CLIP is sensitive to "not."

### 3.2 ALGORITHM DESIGN

Since CLIP lacks the ability to comprehend the concept of "not," using the original "this is not a photo of" as negative prompts do not have the desired effect. Next, we introduce our strategy on how to learn effective negative prompts to help model to understand what is "not".

**Preliminaries: visual prompt learning.** For pre-trained vision-language models, the text input, known as prompt, plays a key role in downstream tasks. However, identifying the right prompt is a non-trivial task, which often takes a significant amount of time for word tuning. To bypass it, in CoOp (Zhou et al., 2022b), the authors propose to learn prompt representations automatically. The learnable prompt $\boldsymbol{V}(l_c)$ for a given class $l_c$ is designed with the following form:

$$\boldsymbol{V}(l_c) = [\boldsymbol{v}_1, \boldsymbol{v}_2 \ldots \boldsymbol{v}_L, \boldsymbol{w}_c],\tag{2}$$

where $\boldsymbol{v}_1, \boldsymbol{v}_2 \ldots \boldsymbol{v}_L$ are learned vectors and can be analog to the context of the hand-designed prompt such as "a photo of a," each $\boldsymbol{v}_i$ has the same dimension as word embeddings (i.e., 512 for CLIP), and $\boldsymbol{w}_c$ is the class token of base class $l_c$. The learned prompt context $[\boldsymbol{v}_1, \ldots \boldsymbol{v}_L]$ can be shared across classes, or it can be class-specific. In our experiments, positive prompts are designed to be shared among classes, and negative prompts are designed to be class-specific. To improve generalization performance, CoCoOp (Zhou et al., 2022a) further learns a Meta-Net built with a two-layer bottleneck structure (Linear-ReLU-Linear) to generate for each image an input-conditional token $m(\mathbf{x})$. Then

the token is added to each context token $\boldsymbol{v}_i(\mathbf{x}) = \boldsymbol{v}_i + m(\mathbf{x})$ for $i \in 1, 2, , L$. To learn prompts, the prediction probability to any ID training sample $(\mathbf{x}_i, y_i) \in T$ is computed as follows:

$$p_i = \frac{\exp\left(\cos(\boldsymbol{f}(\mathbf{x}_i), \boldsymbol{g} \circ \boldsymbol{V}(y_i))/\tau\right)}{\sum_{l_c \in \mathcal{Y}} \exp\left(\cos\left(\boldsymbol{f}(\mathbf{x}_i), \boldsymbol{g} \circ \boldsymbol{V}(l_c)\right)/\tau\right)}, \tag{3}$$

where $\boldsymbol{f}$ and $\boldsymbol{g}$ represent the image and text encoders, respectively. $\tau$ is a temperature parameter, and $\cos(\cdot, \cdot)$ denotes the cosine similarity. The cross-entropy loss is applied to optimize learnable prompt $\boldsymbol{V}$ while keeping the CLIP image and text encoders frozen:

$$\mathcal{L}^+ = -\mathbb{E}_{(\mathbf{x}_i, y_i) \in T} \log p_i. \tag{4}$$

**Negative classifier.** Many OOD detection methods identify OOD samples by analyzing the properties of predictions of the neural networks trained on ID samples (such neural networks trained in a conventional way are called positive classifiers). Thus, the performance of such methods heavily relies on the quality of the features extracted from the well-trained model. However, due to the inertia of neural networks, the features extracted by the model, which is trained only on ID samples, are often inadequate. To alleviate this problem, we propose to learn a negative classifier for each ID class to mine the negative features. To be more specific, for $c$-th negative classifier, it needs to mine the general negative features that samples from class $l_c$ don't have but samples from all other classes have. Thus, the $c$-th negative classifier will produce low activation to samples from class $l_c$ and produce high activation to other classes. By learning additional negative classifiers, we allow models to make decisions from both sides based on different features. A toy example is described below:

Consider a triple classification task with two ID samples per class (details are shown in Figure 5):

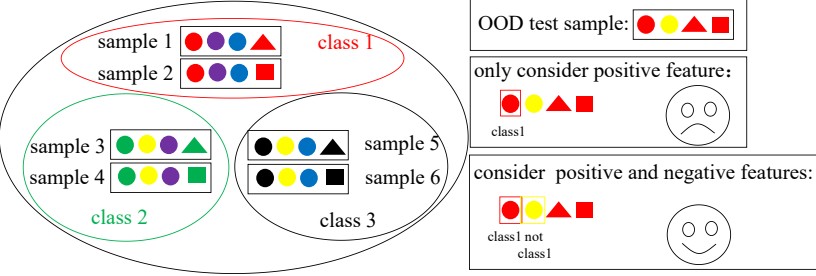

Figure 5: A toy example of a negative classifier. Each geometric figure represents a feature.

When performing a triple classification task, it is clear that the model can easily use ● as the representative feature of "class 1", ● as the representative feature of "class 2", and ● as the representative feature of "class 3". When dealing with an OOD sample with descriptive features in Figure 5, the model trained by the triple classification task can easily classify it as "class 1" with high confidence because the model focuses on the learned feature ● and ignores the other anomalous features. However, when performing a negative classification task, ● will be used to represent "not class 1", ● will be used to represent "not class 2", ● will be used to represent "not class 3". When we add the additional negative classifier, the negative classifier focuses on another feature ● and thus judges it as "not class 1". The positive classifier judges it as "class 1," while the negative classifier judges it as "not class 1", thus reducing the confidence of the model's prediction of the OOD sample.

**Negative prompt learning.** As described above, we want to learn negative prompts to capture negative features. These negative prompts construct negative classifiers, thus the powerful visual features extracted by clip can be fully utilized. According to the example in Figure 5, such negative prompts $\bar{\boldsymbol{V}}$ should satisfy the following two properties:

- The negative prompt representation $\bar{\boldsymbol{V}}(l_c) = [\boldsymbol{v}_1, \boldsymbol{v}_2 \ldots \boldsymbol{v}_L, \boldsymbol{w}_c]$ for given class $l_c$ should produce a low match with images whose label is $l_c$, as it represents negation.
- The negative prompt representation $\bar{\boldsymbol{V}}(l_c) = [\boldsymbol{v}_1, \boldsymbol{v}_2 \ldots \boldsymbol{v}_L, \boldsymbol{w}_c]$ for given class $l_c$ should produce a high match with all images whose labels are not $l_c$, as "this is not a photo of a [CLASS]" is the correct description for all images whose labels are not "[CLASS]"

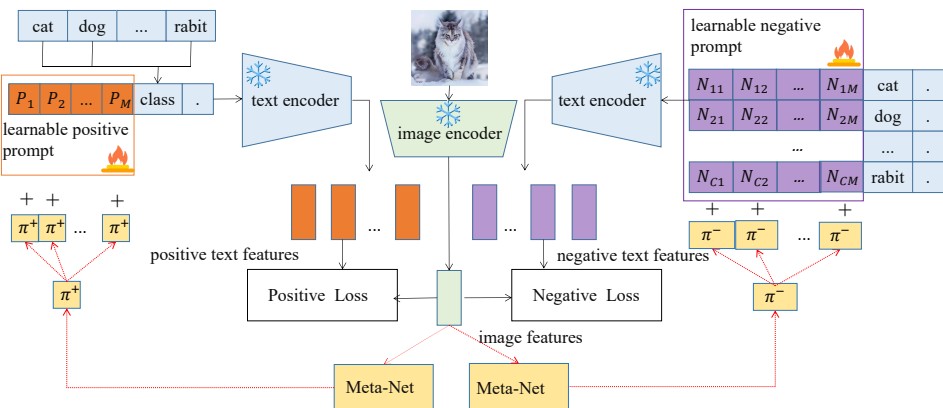

Figure 6: An overview of LSN. The CLIP text encoder and image encoder are fixed. We optimize positive prompts and negative prompts by Eq. equation 4 and Eq. equation 8 respectively. When we use CoOp to learn prompts, we don't use the Meta-Net.

Based on the above two properties, we design a novel loss $\mathcal{L}$ to optimize the negative prompts:

$$\mathcal{L} = -\mathbb{E}_{(\mathbf{x}_i, y_i) \in T} \log p_i^-, \tag{5}$$

where

$$p_i^- = \frac{\exp\left(1/\tau - \cos(\boldsymbol{f}(\mathbf{x}_i), \boldsymbol{g} \circ \bar{\boldsymbol{V}}(y_i))/\tau\right)}{\sum_{l_c \in \mathcal{Y}} \exp\left(1/\tau - \cos\left(\boldsymbol{f}(\mathbf{x}_i), \boldsymbol{g} \circ \bar{\boldsymbol{V}}(l_c)\right)/\tau\right)}. \tag{6}$$

In positive prompt learning, most of the positive features of a class are carried by the class names, and the learned positive prompt only serves as a calibration, so learning a unified positive prompt for all classes is sufficient. However, in negative prompt learning, the situation is quite different. The negative features of a class cannot be carried by the class name but should be included in the learned negative prompts, and the negative features of a class are often diverse. Therefore, in negative prompt learning, we learn class-specific prompts, and for each class, we learn a set of negative prompts. We encourage the learned negative prompts to be different by the following semantic orthogonality loss:

$$\mathcal{L}_{reg} = \sum_{c=1}^{C} \sum_{i=1}^{K} \sum_{j=i+1}^{K} |cos(t_c^i, t_c^j)|, \tag{7}$$

where $t_c^i = \boldsymbol{g} \circ \bar{\boldsymbol{V}}^i(l_c)$ is the text feature of $i$-th negative prompts for class $l_c$. For each class, we learn $K$ different negative prompts to attend to different negative features. The overall loss function combines the empirical classification loss and regularisation term as :

$$\mathcal{L}^- = \mathcal{L} + \lambda \mathcal{L}_{reg}, \tag{8}$$

where $\lambda$ is a hyper-parameter. An overview of our method is shown in Figure 6. And for testing, we use the difference of MCM score (Ming et al., 2022) between positive prompts and negative prompts as the score function: given a sample x, the score is

$$S(\mathbf{x}) = \max_c \frac{\exp\left(\cos\left(\boldsymbol{f}(\mathbf{x}), \boldsymbol{g} \circ \boldsymbol{V}(l_c)\right)/\tau\right)}{\sum_{j'=1}^{C} \exp\left(\cos\left(\boldsymbol{f}(\mathbf{x}), \boldsymbol{g} \circ \boldsymbol{V}(l_j)\right)/\tau\right)} - \min_c \frac{\exp\left(\cos\left(\boldsymbol{f}(\mathbf{x}), \boldsymbol{g} \circ \bar{\boldsymbol{V}}(l_c)\right)/\tau\right)}{\sum_{j'=1}^{C} \exp\left(\cos\left(\boldsymbol{f}(\mathbf{x}), \boldsymbol{g} \circ \bar{\boldsymbol{V}}(l_j)\right)/\tau\right)}. \tag{9}$$

Leveraging the designed score $S(\mathbf{x})$, we can endow OOD detection with the ability to say "no".

## 4 EXPERIMENTS

### 4.1 DATASETS

In order to fully validate the effectiveness of our proposed method, we conduct experiments on both large-scale and small-scale datasets. For the large-scale dataset, we use 100 randomly selected categories from ImageNet-1k (Deng et al., 2009) as ID dataset, following the MCM (Ming et al., 2022), and the selected classes are kept consistent with MCM. For the OOD test datasets, we

Table 1: OOD detection performance on ImageNet-100 as ID. Values are percentages. **Bold** numbers are superior results. ↑ indicates larger values are better, and ↓ indicates smaller values are better. ∗ denotes the results obtained from the relevant paper, and ⋆ denotes the results reproduced by ourself.

| Methods | OOD Dataset | | | | | | | | Average | | ID ACC |
| | iNaturalist | | SUN | | Places | | Textures | | | | |
| | FPR95↓ | AUROC↑ | FPR95↓ | AUROC↑ | FPR95↓ | AUROC↑ | FPR95↓ | AUROC↑ | FPR95↓ | AUROC↑ | |
|---|---|---|---|---|---|---|---|---|---|---|---|
| MCM∗ | 18.13 | 96.77 | 36.45 | 94.54 | 34.52 | 94.36 | 41.22 | 92.25 | 32.58 | 94.48 | 87.88 |
| MSP⋆ | 23.55 | 95.92 | 37.02 | 92.45 | 40.76 | 91.23 | 24.40 | 94.90 | 31.43 | 93.63 | 91.93 |
| Vim⋆ | 20.11 | 96.22 | 38.56 | 93.12 | 44.01 | 87.33 | 33.12 | 93.24 | 33.95 | 92.48 | 91.93 |
| VOS⋆ | 12.55 | 97.53 | 39.65 | 92.78 | 38.84 | 92.89 | 15.27 | 97.19 | 26.58 | 95.10 | 91.87 |
| NPOS⋆ | 9.56 | 97.94 | 14.93 | 97.04 | 17.51 | 96.33 | 7.59 | 98.07 | 12.40 | 97.35 | 91.89 |
| CoOp⋆ | 9.30 | 97.95 | 11.64 | 97.61 | 17.45 | 96.53 | 15.94 | 96.90 | 13.58 | 97.25 | 92.88 |
| CoCoOp⋆ | 11.76 | 97.84 | 14.28 | 97.13 | 15.16 | 96.73 | 18.27 | 96.54 | 14.86 | 97.06 | 92.24 |
| CoOp + LSN (ours) | 5.74 | 98.32 | 12.42 | 97.53 | 14.62 | 96.89 | 9.17 | 97.82 | **10.49** | **97.64** | 92.88 |
| CoCoOp + LSN (ours) | 4.93 | 98.92 | 8.23 | 97.98 | 12.82 | 97.19 | 8.26 | 98.11 | **8.56** | **98.05** | 92.24 |

use iNaturalist (Horn et al., 2018), SUN (Xiao et al., 2010), PLACES (Zhou et al., 2018), and TEXTURE (Cimpoi et al., 2014) following MCM. For completeness, we also conduct experiments on the full ImageNet-1k dataset. For the small-scale dataset, following ZOC (Esmaeilpour et al., 2022), we evaluate the performance of our method on splits of CIFAR10 (Krizhevsky et al., 2009), CIFAR100 (Krizhevsky et al., 2009), CIFAR+10, CIFAR+50 and TinyImagenet (Le & Yang, 2015). For CIFAR+10, 4 non-animal classes in CIFAR10 are selected as ID classes, and 10 animal classes in CIFAR100 are selected as OOD classes. For CIFAR+50, 4 non-animal classes in CIFAR10 are selected as ID classes, and 50 animal classes in CIFAR100 are selected as OOD classes.

## 4.2 MODEL ARCHITECTURE AND IMPLEMENTATION DETAILS

**Experimental setup.** In experiments, we mainly used CLIP as the pre-trained model. For ImageNet-100 and ImageNet-1k benchmarks, following the same setup in MCM (Ming et al., 2022), we use CLIP-B/16 as the backbone model. For CIFAR10 benchmark, in agreement with ZOC, we use CLIP-B/32. All the images are resized to size $224 * 224$. For some baselines that do not require a text encoder, we use CLIP's image encoder as their backbone. For positive prompt learning, we closely follow the original implementations in CoOp and CoCoOp. And for negative prompt learning, we use Adam as optimizer (Kingma & Ba, 2015) for faster convergence. The initial learning rate is set to 1e-3. For each class, we learn three sets of negative prompts, and $\lambda$ is set to 0.1.

**Evaluation Metrics.** For evaluation, we use the following metrics: (1) the false positive rate (FPR95) of OOD samples when the true positive rate of in-distribution samples is at 95%, (2) the area under the receiver operating characteristic curve (AUROC), and (3) ID classification accuracy (ID ACC). When determining the classes of ID samples, we only use positive prompts.

## 4.3 BASELINE METHODS

We compare LSN with several baseline methods as follows, MCM (Ming et al., 2022), MSP (Fort et al., 2021), ODIN (Liang et al., 2018), Energy (Liu et al., 2020), GradNorm (Huang et al., 2021), Vim (Wang et al., 2022a), KNN (Sun et al., 2022b), VOS (Du et al., 2022), NPOS (Tao et al., 2023), ZOC (Esmaeilpour et al., 2022) and CLIPN(Wang et al., 2023a). We also compared LSN with vanilla prompt learning methods such as CoOp (Zhou et al., 2022b) and CoCoOp (Zhou et al., 2022a).

## 4.4 MAIN RESULTS

Table 1, Table 2 and Table 3 exhibit evaluations of our method on several datasets. Results show that LSN achieves the highest average performance compared to other baselines. Compared to the OOD detection baseline MCM for the vision-language model, our method LSN reduces the FPR95 from 32.58% to 8.56% on the ImageNet-100 dataset, a great improvement of 24.02%. Even compared to a recent very strong baseline work NPOS, our method still achieves higher performance. Please note that NPOS needs to fine-tune CLIP using all the training samples and needs to use synthetic outliers, while our method only needs to use part of data to train prompts. Even so, our approach still beats NPOS. On the large-scale dataset ImagtNet-1k, compared to MCM, LSN also reduces the FPR95 from 43.55% to 30.22%, a significant improvement of 13.33%. Compared to CLIPN, LSN achieves consistent performance at a smaller computational cost. On the small-scale datasets CIFAR10, CIFAR100, CIFAR+10, CIFAR+100, and TinyImagenet, LSN also outperforms previous baseline works. The superior performance shows that the learned positive prompts and negative

Table 2: OOD detection performance on ImageNet-1k as ID.

| Methods | OOD Dataset | | | | | | | | | | ID ACC |
|---|---|---|---|---|---|---|---|---|---|---|---|
| | iNaturalist | | SUN | | Places | | Textures | | Average | | |
| | FPR95↓ | AUROC↑ | FPR95↓ | AUROC↑ | FPR95↓ | AUROC↑ | FPR95↓ | AUROC↑ | FPR95↓ | AUROC↑ | |
| MCM* | 32.08 | 94.41 | 39.21 | 92.28 | 44.88 | 89.83 | 58.05 | 85.96 | 43.55 | 90.62 | 68.53 |
| MSP* | 54.05 | 87.43 | 73.37 | 78.03 | 72.98 | 78.03 | 68.85 | 79.06 | 67.31 | 80.64 | 79.64 |
| ODIN* | 30.22 | 94.65 | 54.04 | 87.17 | 55.06 | 85.54 | 51.67 | 87.85 | 47.75 | 88.80 | 79.64 |
| Energy* | 29.75 | 94.68 | 53.18 | 87.33 | 56.40 | 85.60 | 51.35 | 88.00 | 47.67 | 88.90 | 79.64 |
| GradNorm* | 81.50 | 72.56 | 82.00 | 72.86 | 80.41 | 73.70 | 79.36 | 70.26 | 80.82 | 72.35 | 79.64 |
| Vim* | 32.19 | 93.16 | 54.01 | 87.19 | 60.67 | 83.75 | 53.94 | 87.18 | 50.20 | 87.82 | 79.64 |
| KNN* | 29.17 | 94.52 | 35.62 | 92.67 | 39.61 | 91.02 | 64.35 | 85.67 | 42.19 | 90.97 | 79.64 |
| VOS* | 31.65 | 94.53 | 43.03 | 91.92 | 41.62 | 90.23 | 56.67 | 86.74 | 43.24 | 90.86 | 79.64 |
| VOS+* | 28.99 | 94.62 | 36.88 | 92.57 | 38.39 | 91.23 | 61.02 | 86.33 | 41.32 | 91.19 | 79.58 |
| NPOS* | 16.58 | 96.19 | 43.77 | 90.44 | 45.27 | 89.44 | 46.12 | 88.80 | 37.93 | 91.22 | 79.42 |
| CoOp* | 29.47 | 94.89 | 31.34 | 93.36 | 40.28 | 90.07 | 54.25 | 87.58 | 38.83 | 91.47 | 72.93 |
| CoCoOp* | 30.74 | 94.73 | 31.18 | 93.15 | 38.75 | 90.63 | 53.84 | 87.92 | 38.63 | 91.61 | 71.89 |
| CLIPN* | 23.94 | 95.27 | 26.17 | 93.93 | 33.45 | 92.28 | 40.83 | 90.93 | 31.10 | **93.10** | 68.53 |
| CoOp + LSN (ours) | 23.48 | 95.47 | 29.84 | 93.45 | 36.43 | 90.87 | 38.16 | 89.52 | 31.97 | 92.33 | 72.93 |
| CoCoOp + LSN (ours) | 21.56 | 95.83 | 26.32 | 94.35 | 34.48 | 91.25 | 38.54 | 90.42 | **30.22** | 92.96 | 71.89 |

Table 3: OOD detection performance in AUROC on CIFAR10, CIFAR100, CIFAR+10, CIFAR+50 and TinyImagenet. Each result in the table is the average of 5 splits of each dataset. We do not show the FPR95 score for ZOC as it is not reported in the original paper.

| Methods | Dataset | | | | | | | | | |
|---|---|---|---|---|---|---|---|---|---|---|
| | CIFAR10 | | CIFAR100 | | CIFAR+10 | | CIFAR+50 | | TinyImagenet | |
| | FPR95↓ | AUROC↑ | FPR95↓ | AUROC↑ | FPR95↓ | AUROC↑ | FPR95↓ | AUROC↑ | FPR95↓ | AUROC↑ |
| MCM* | 51.87 | 90.06 | 79.18 | 82.68 | 46.68 | 94.03 | 46.66 | 94.20 | 74.36 | 84.60 |
| ZOC* | - | 93.00 | - | 82.10 | - | 97.80 | - | 97.60 | - | 84.60 |
| MSP* | 50.31 | 91.63 | 72.39 | 82.59 | 12.82 | 97.32 | 15.60 | 96.54 | 68.00 | 84.15 |
| CoOp | 35.21 | 94.44 | 72.24 | 84.67 | 14.26 | 97.36 | 26.37 | 95.86 | 63.24 | 87.84 |
| CoCoOp | 38.16 | 93.82 | 69.38 | 85.03 | 11.69 | 97.52 | 20.57 | 96.62 | 66.56 | 87.42 |
| CoOp + LSN (ours) | **25.16** | **95.23** | **53.28** | **86.14** | **6.27** | **98.75** | **8.25** | **98.21** | **53.27** | **88.15** |
| CoCoOp + LSN (ours) | **27.63** | **95.17** | **51.27** | **86.32** | **5.92** | **98.93** | **7.65** | **98.32** | **56.83** | **88.32** |

prompts greatly improve the out-of-distribution detection capability of the model, demonstrating the effectiveness of the proposed method.

## 4.5 EMPIRICAL ANALYSIS

In this section, we conduct experiments on ablation study. Unless otherwise noted, experiments are conducted on the ImageNet-100 dataset with the model of CLIP-B/16 and the method of CoCoOp.

**The effect of positive prompts and negative prompts.** We conduct ablation studies to examine the effectiveness of learned positive prompts and negative prompts, respectively. Experimental results show that both learned positive prompts and negative prompts improve the out-of-distribution detection ability of the model, as shown in Table 4. The proposed semantic orthogonality loss also contributes to performance improvement. In addition to this, we conduct a comparison experiment of learning multiple (three) positive prompts. However, experimental results show that learning multiple positive prompts does not further improve performance, while learning negative prompts simultaneously does, further demonstrating the necessity of learning negative prompts.

**The gap between positive prompt learning and negative prompt learning.** Although our approach to learning negative prompts looks basically the same as CoOp and CoCoOp, there is a fundamental difference between positive prompt learning and negative prompt learning. In positive prompt learning, using a shared prompt across classes is enough to achieve good performance. This is because in positive prompt learning, the positive features of each class are carried by the class names, and the positive prompt is only used to calibrate these features to the downstream dataset. In negative prompt learning, on the other hand, the situation is completely different. The negative features need to be contained in the learned negative prompts, and the class names appear to be less important. As shown in Table 5, for positive prompt learning, there is little performance difference between using class-shared prompts and class-specific prompts, while when class names are not used, the performance degradation is more clear. In negative prompt learning, using class-specific prompts can lead to large performance gains, while the class names are not important.

Table 4: Both positive and negative prompts can improve the OOD detection ability.

| Methods | OOD Dataset | | | | | | | | | |
| | iNaturalist | | SUN | | Places | | Textures | | Average | |
| | FPR95↓ | AUROC↑ | FPR95↓ | AUROC↑ | FPR95↓ | AUROC↑ | FPR95↓ | AUROC↑ | FPR95↓ | AUROC↑ |
|---|---|---|---|---|---|---|---|---|---|---|
| MCM | 18.13 | 96.77 | 36.45 | 94.54 | 34.52 | 94.36 | 41.22 | 92.25 | 32.58 | 94.48 |
| full model | 4.93 | 98.92 | 8.23 | 97.98 | 12.82 | 97.19 | 8.26 | 98.11 | 8.56 | 98.05 |
| w/o positive prompts | 5.16 | 98.31 | 20.48 | 95.53 | 20.92 | 95.58 | 11.77 | 97.52 | 14.58 | 96.73 |
| w/o negative prompts | 11.76 | 97.84 | 14.28 | 97.13 | 15.16 | 96.73 | 18.27 | 96.54 | 14.86 | 97.06 |
| w/o semantic orthogonality loss | 5.01 | 98.48 | 10.13 | 97.68 | 13.31 | 96.85 | 14.47 | 97.89 | 10.73 | 97.73 |
| multi positive prompts | 11.37 | 97.90 | 15.24 | 97.09 | 16.45 | 96.58 | 17.79 | 96.62 | 15.21 | 97.04 |

Table 5: The gap between positive prompt learning and negative prompt learning.

| Methods | OOD Dataset | | | | | | | | | |
| | iNaturalist | | SUN | | Places | | Textures | | Average | |
| | FPR95↓ | AUROC↑ | FPR95↓ | AUROC↑ | FPR95↓ | AUROC↑ | FPR95↓ | AUROC↑ | FPR95↓ | AUROC↑ |
|---|---|---|---|---|---|---|---|---|---|---|
| pos-specific-w name | 11.65 | 97.66 | 16.27 | 96.84 | 17.92 | 96.62 | 19.63 | 96.42 | 16.61 | 96.89 |
| pos-shared-w name | 11.76 | 97.84 | 14.28 | 97.13 | 15.16 | 96.73 | 18.27 | 96.54 | 14.86 | 97.06 |
| pos-specific-w/o name | 21.83 | 96.62 | 26.72 | 95.49 | 22.49 | 96.15 | 21.82 | 96.24 | 23.22 | 96.12 |
| neg-specific-w name | 13.46 | 97.14 | 28.99 | 94.01 | 29.44 | 93.94 | 15.88 | 96.79 | 21.94 | 95.47 |
| neg-shared-w name | 81.78 | 83.62 | 86.21 | 77.95 | 85.45 | 78.69 | 71.64 | 88.83 | 81.27 | 82.27 |
| neg-specific-w/o name | 12.86 | 97.25 | 30.01 | 93.39 | 28.79 | 94.44 | 18.29 | 96.65 | 22.48 | 95.43 |

# 5 RELATED WORK

**Out-of-distribution detection.** The goal of OOD detection is to enable the model to discriminate between ID samples and OOD samples while maintaining classification accuracy on ID samples. The representative OOD detection methods mainly include: Post-hoc Detection (Hendrycks & Gimpel, 2017) (Liu et al., 2020) (Park et al., 2023), Confidence Enhancement Methods (Hein et al., 2019) (Bitterwolf et al., 2020) (Wang et al., 2022b) and Outlier Exposure (Hendrycks et al., 2019) (Wang et al., 2023b). Among them, post-hoc methods have the advantage of being easy to use without modifying the training procedure and objective. Recently, inspired by the success of pre-trained vision-language models, some works have enriched OOD detection from a single-modal to a multi-modal regime. Ming *et al.* (Ming et al., 2022) proposes maximum concept matching to align visual features with textual concepts. (Ming & Li, 2023) further investigates the effect of fine-tuning on OOD detection in large vision-language models, and the MCM score is highlighted as effective. These CLIP-based OOD detection methods deliver superior performance in a simple way.

**Prompt learning.** In recent years, large vision-language models (VLMs) such as CLIP (Radford et al., 2021) have demonstrated surprising results in zero-shot and few-shot learning tasks. When applied to downstream tasks, the performance of VLMs can be greatly affected by the prompts. Task-specific prompts can significantly improve performance, but require laborious prompt engineering. To this end, inspired by prompt learning in language tasks, a series of prompt learning methods (Zhou et al., 2022b; Lu et al., 2022) in computer vision are proposed. Unlike the above methods, we aim to improve the out-of-distribution detection capability of the model through learning both positive and negative prompts, thus fully leveraging the capabilities of CLIP for OOD detection.

**Complementary label learning.** Complementary Label Learning (CLL) is a new problem in weakly supervised learning that allows models to learn from complementary labels. (Ishida et al., 2017) proposes an unbiased risk estimator (URE) within several specific multi-class loss functions to the classification risk and theoretically establishes an estimation error bound. (Yu et al., 2018) consideres the biased CLs. (Feng et al., 2020) derives an URE of the ordinary risk for multiple complementary labels, and improves it by minimizing properly chosen upper bounds. LSN takes all ID labels except the correct label as complementary labels.

# 6 CONCLUSION

In this paper, we propose negative classifiers to accurately identify when an image does not belong to a particular category. We discuss that the features learned by the negative classifier can be an effective complement to the features learned by the positive classifier. Further, instead of learning such negative classifiers from scratch, we construct them on top of CLIP. With the help of CLIP, we build such negative classifiers by learning negative prompts. Extensive experiments on standard benchmarks indicate the effectiveness of the proposed method LSN.

## ACKNOWLEDGMENTS

This work was supported in part by NSFC No. 62222117, the Fundamental Research Funds for the Central Universities under contract WK3490000005, and KY2100000117. YGZ and BH were supported by the NSFC General Program No. 62376235, Guangdong Basic and Applied Basic Research Foundation No. 2022A1515011652, HKBU Faculty Niche Research Areas No. RC-FNRA-IG/22-23/SCI/04, and HKBU CSD Departmental Incentive Scheme. TL is partially supported by the following Australian Research Council projects: FT220100318, DP220102121, LP220100527, LP220200949, and IC190100031.

## ETHIC STATEMENT

This paper does not raise any ethical concerns. This study does not involve any human subjects, practices to data set releases, potentially harmful insights, methodologies and applications, potential conflicts of interest and sponsorship, discrimination/bias/fairness concerns, privacy and security issues, legal compliance, and research integrity issues.

## REPRODUCIBILITY STATEMENT

We summarize our efforts below to facilitate reproducible results:

- **Methodology.** Our method is fully documented in Section 3.2 with the pseudo algorithm detailed in Algorithm 1. Hyperparameters are specified in Section 4.2, with a thorough ablation study provided in Section 4.5.
- **Datasets.** We use publicly available datasets, which are described in detail in Section 4.2 and Appendix C.
- **Open Source.** Code is available at https://github.com/junz-debug/lsn.

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

APPENDIX

## A  LIMITATIONS

- LSN relies heavily on the features learned by CLIP. If the features extracted by CLIP itself for some categories of images are not strongly discriminative, the effect of learning the prompts based on these features may be poor.
- Compared to CoOp, LSN doubles the training and inference time.
- LSN uses only a small number of samples to learn the prompts to improve the model's OOD detection. However, limited by the small number of learnable parameters, the model's detection ability is not improved more when increasing the training samples further, which results in LSN not benefiting from more training samples.

## B  ALGORITHM OF LSN

We summarize our algorithm in implementation, as shown in Algorithm 1.

---
**Algorithm 1** Learn to Say No (LSN)

---
**Input:** Pre-trained VLM encoders for image $f$ and text $g$; ID training samples $T$; ID labels $\mathcal{Y}$
**Output:** Score function $S$ and OOD predictor $G$
1: Randomly initialize the positive prompts $V$ and negative prompts $\bar{V}$
2: **for** Epoch=1 : $E$ **do**
3:     Sample a mini-batch $\{(\tilde{\mathbf{x}}_1, \tilde{y}_1), ..., (\tilde{\mathbf{x}}_b, \tilde{y}_b)\}$ from ID training samples $I$
4:     Compute image features $f(\tilde{\mathbf{x}}_i), i = 1, \ldots, b$
5:     Compute positive text embeddings $g \circ V(l_c), c = 1, .., C$
6:     Compute negative text embeddings $g \circ \bar{V}(l_c), c = 1, .., C$
7:     Compute positive loss $\mathcal{L}^+$ using Eq.equation 4 and update $V$ by gradient descent
8:     Compute negative loss $\mathcal{L}^-$ using Eq.equation 8 and update $\bar{V}$ by gradient descent
9: Compute the score function by Eq.equation 9 and obtain OOD predictor by Eq. equation 1

---

## C  DETAILS OF DATASETS

In this section, we provide dataset details.

### C.0.1  IMAGENET BENCHMARK

Following MCM (Ming et al., 2022), we choose 100 classes from ImageNet-1k (Deng et al., 2009) to create ImageNet-100. The chosen classes are the same as MCM:

n03877845, n03000684, n03110669, n03710721, n02825657, n02113186, n01817953, n04239074, n02002556, n04356056, n03187595, n03355925, n03125729, n02058221, n01580077, n03016953, n02843684, n04371430, n01944390, n03887697, n04037443, n02493793, n01518878, n03840681, n04179913, n01871265, n03866082, n03180011, n01910747, n03388549, n03908714, n01855032, n02134084, n03400231, n04483307, n03721384, n02033041, n01775062, n02808304, n13052670, n01601694, n04136333, n03272562, n03895866, n03995372, n06785654, n02111889, n03447721, n03666591, n04376876, n03929855, n02128757, n02326432, n07614500, n01695060, n02484975, n02105412, n04090263, n03127925, n04550184, n04606251, n02488702, n03404251, n03633091,

n02091635, n03457902, n02233338, n02483362, n04461696, n02871525, n01689811, n01498041, n02107312, n01632458, n03394916, n04147183, n04418357, n03218198, n01917289, n02102318, n02088364, n09835506, n02095570, n03982430, n04041544, n04562935, n03933933, n01843065, n02128925, n02480495, n03425413, n03935335, n02971356, n02124075, n07714571, n03133878, n02097130, n02113799, n09399592, n03594945.

Similarly, we use subsets from iNaturalist (Horn et al., 2018), SUN (Xiao et al., 2010), Places (Zhou et al., 2018) and Texture (Cimpoi et al., 2014) as OOD datasets, which are created by Huang *et al.* (**?**). The classes from OOD datasets do not overlap with ImageNet-1k. A brief description of them is as follows:

**iNaturalist** contains images from the natural world. It has 13 super-categories and 5089 sub-categories covering plants, insects, birds, mammals, and so on. The subset containing 110 plant classes not showing in ImageNet-1k is chosen as the OOD test set.

**SUN** contains 899 categories that cover more than indoor, urban, and natural places. We use the subset that contains 50 natural objects that do not overlap with ImageNet-1k.

**Places** contains photos labeled with scene semantic categories from three macro-classes: Indoor, Nature, and Urban. We use a subset sampled from 50 categories that are not present in ImageNet-1k.

**Texture** contains images of textures and abstracted patterns. As no categories overlap with ImageNet-1k, we use the entire dataset.

### C.0.2 CIFAR10 BENCHMARK

Following ZOC (Esmaeilpour et al., 2022), we also evaluate our methods LSN on CI-FAR10 (Krizhevsky et al., 2009), CIFAR100 (Krizhevsky et al., 2009), CIFAR+10 (Krizhevsky et al., 2009), CIFAR+50 (Krizhevsky et al., 2009) and TinyImagenet datasets (Krizhevsky et al., 2009). For CIFAR10, 6 classes are selected as ID classes, and the 4 remaining classes are selected as OOD classes. Experiments are conducted on 5 different splits. For CIFAR+10, 4 non-animal classes from CIFAR10 are selected as ID classes, and 10 animal classes from CIFAR100 are selected as OOD classes. For CIFAR+50, 4 non-animal classes from CIFAR10 are selected as ID classes, and all 50 animal classes from CIFAR100 are selected as OOD classes. For CIFAR100, 20 consecutive classes are selected as ID classes. The remaining 80 classes are selected as OOD classes. For TinyImagenet, 20 classes are used as ID classes, and the remaining 180 classes are used as OOD classes. All the split is the same as ZOC. A more detailed split information is shown in Table 6.

## D EXPERIMENTAL DETAILS

In this section, we provide the implementation details.

### D.0.1 SOFTWARE AND HARDWARE

We use Python 3.8.16 and Pytorch 1.12.1 and several NVIDIA GeForce RTX 2080, 3090, and 4090 GPUs.

### D.0.2 TRAINING DETAILS

As mentioned in the formal paper, we learn both positive prompts and negative prompts. For positive prompts, we use unified context, and for negative prompts, we use class-specific context. As the CLIP image encoder and text encoder are frozen in experiments, we extract images' features in advance to speed up the training process rather than using original images. For positive prompts, training is done with SGD and an initial learning rate of $0.002$. The scaling factor $\tau$ is set to $0.01$. When we use CoOp, the prompt length is set to 16, the max epoch is set to 50, and the batch size is set to 8. When we use CoCoOp, the prompt length is set to 4, the max epoch is set to 10, and the batch size is set to 1. For negative prompts, training is done with Adam, and an initial learning rate of $0.001$. The prompt length is set to 16. The scaling factor $\tau$ is set to $0.05$. When we use CoOp, the max epoch is set to 100, and batch size is set to 8. When we use CoCoOp, the max epoch is set to 10, and batch size is set to 1. As the learned negative prompts benefit from a large number of training samples, to balance

Table 6: Data splits for CIFAR10 benchmark. The numbers in the table represent the class indices for ID classes except CIFAR+ cases. For CIFAR+ experiments, we provide OOD classes since non-animal classes are utilized for ID classes. For CIFAR+50, there is only one split, and for the sake of the table's completeness, we have repeated it here 5 times.

| | split 1 | split 2 | split 3 | split 4 | split 5 |
|---|---|---|---|---|---|
| CIFAR10 | 0, 1, 2, 4, 5, 9 | 0, 3, 5, 7, 8, 9 | 0, 1, 5, 6, 7, 8 | 3, 4, 5, 7, 8, 9 | 0, 1, 2, 3, 7, 8 |
| CIFAR100 | 0,1,2,3, 4,5,6,7, 8,9,10,11, 12,13,14,15, 16,17,18,19 | 20,21,22,23, 24,25,26,27, 28,29,30,31, 32,33,34,35, 36,37,38,39 | 40,41,42,43, 44,45,46,47, 48,49,50,51, 52,53,54,55, 56,57,58,59 | 60,61,62,63, 64,65,66,67, 68,69,70,71, 72,73,74,75, 76,77,78,79 | 80,81,82,83, 84,85,86,87, 88,89,90,91, 92,93,94,95, 96,97,98,99 |
| CIFAR+10 | 31,65,45,98,77, 93,34,63,26,44 | 11,97,98,75,66, 77,95,99,93,7 | 11,24,15,95,93, 34,32,88,63,2 | 11,1,64,42,45, 38,66,67,63,44 | 19,42,21,78,98, 72,15,46,66,3 |
| CIFAR+50 | 19,29,11,1,31, 97,80,74,64,42, 65,21,24,78,45, 73,14,6,98,36, 55,72,43,35,27, 50,15,18,46,75, 38,66,77,95,99, 93,4,34,32,88, 67,30,63,26,79, 44,7,2,91,3 | 19,29,11,1,31, 97,80,74,64,42, 65,21,24,78,45, 73,14,6,98,36, 55,72,43,35,27, 50,15,18,46,75, 38,66,77,95,99, 93,4,34,32,88, 67,30,63,26,79, 44,7,2,91,3 | 19,29,11,1,31, 97,80,74,64,42, 97,80,74,64,42, 97,80,74,64,42, 97,80,74,64,42, 97,80,74,64,42, 97,80,74,64,42, 97,80,74,64,42, 97,80,74,64,42, 97,80,74,64,42, | 19,29,11,1,31, 97,80,74,64,42, 97,80,74,64,42, 97,80,74,64,42, 97,80,74,64,42, 97,80,74,64,42, 97,80,74,64,42, 97,80,74,64,42, 97,80,74,64,42, 97,80,74,64,42, | 19,29,11,1,31, 97,80,74,64,42, 97,80,74,64,42, 97,80,74,64,42, 97,80,74,64,42, 97,80,74,64,42, 97,80,74,64,42, 97,80,74,64,42, 97,80,74,64,42, 97,80,74,64,42, |
| TinyImagenet | 192,112,145,107, 91,180,144,193, 10,125,186,28, 72,124,54,77, 157,169,104,166 | 156,157,167,175, 153,11,147,0, 199,171,132,60, 87,190,101,111, 193,71,131,192 | 28,15,103,33, 90,167,61,13, 124,159,49,12, 54,78,82,107, 80,25,140,46 | 128,132,123,72, 154,35,86,10, 188,28,85,89, 91,82,116,65, 96,41,134,25 | 102,79,47,106, 59,93,145,10, 62,175,76,183, 48,130,38,186, 44,8,29,26 |

time cost and performance, on ImageNet-100, CIFAR10, CIFAR100, CIFAR+10, CIFAR+50, and TinyImagenet, we use 128 samples per class. And on ImageNet-1k, we use 64 samples per class.

### D.0.3 TEST DETAILS

For testing, we use the difference of similarities between positive prompts and negative prompts as a score function. In a formal paper, we give an abbreviated form. Here, we describe it in more detail.

Once we finish the training process, we get the positive prompts $V$ and negative prompts $\bar{V}$. Then we get the positive prompt representation $V(l_c)$ and negative prompt representation $\bar{V}(l_c)$ for each class $l_c \in \mathcal{Y}_{\text{in}} = \{l_1, l_2 ... l_C\}$. For any given test sample $\mathbf{x}$, we calculate the positive cosine similarity between the image feature $\boldsymbol{f}(\mathbf{x})$ and the positive text feature $\boldsymbol{g} \circ V(l_c)$:

$$\boldsymbol{s}_c = \cos\left(\boldsymbol{f}(\mathbf{x}), \boldsymbol{g} \circ V(l_c)\right) = \frac{\boldsymbol{f}(\mathbf{x}) \cdot \boldsymbol{g} \circ V(l_c)}{\|\boldsymbol{f}(\mathbf{x})\| \cdot \|\boldsymbol{g} \circ V(l_c)\|}, \tag{10}$$

Following MCM, we calculate the maximum concept matching (MCM) score as follows:

$$S_{\text{MCM}}\left(\mathbf{x}; \mathcal{Y}_{\text{in}}\right) = \max_c \frac{e^{\boldsymbol{s}_c(\mathbf{x})/\tau}}{\sum_{j=1}^{C} e^{\boldsymbol{s}_j(\mathbf{x})/\tau}}. \tag{11}$$

Similarly, we calculate the negative cosine similarity between the image feature $\boldsymbol{f}(\mathbf{x})$ and the negative text feature $\boldsymbol{g} \circ \bar{V}(l_c)$:

$$\bar{\boldsymbol{s}}_c = \cos\left(\boldsymbol{f}(\mathbf{x}), \boldsymbol{g} \circ \bar{V}(l_c)\right) = \frac{\boldsymbol{f}(\mathbf{x}) \cdot \boldsymbol{g} \circ \bar{V}(l_c)}{\|\boldsymbol{f}(\mathbf{x})\| \cdot \|\boldsymbol{g} \circ \bar{V}(l_c)\|}, \tag{12}$$

Table 7: OOD detection performance on ImageNet-100 as ID.

| Methods | iNaturalist | | SUN | | Places | | Textures | | Average | |
|---------|---------|---------|---------|---------|---------|---------|---------|---------|---------|---------|
| | FPR95↓ | AUROC↑ | FPR95↓ | AUROC↑ | FPR95↓ | AUROC↑ | FPR95↓ | AUROC↑ | FPR95↓ | AUROC↑ |
| MCM | 18.02 | 96.98 | 47.99 | 92.67 | 53.63 | 91.12 | 45.19 | 92.02 | 41.21 | 93.19 |
| DualCoOp | 15.98 | 97.02 | 24.67 | 95.71 | 36.05 | 93.11 | 17.18 | 96.72 | 23.47 | 95.64 |
| LSN | 13.59 | 97.67 | 17.85 | 96.32 | 26.66 | 94.26 | 20.72 | 96.65 | 19.70 | 96.22 |

Then we calculate the minimum concept matching (MinCM) score as follows:

$$S_{\mathrm{MinCM}}(\mathbf{x}; \mathcal{Y}_{\mathrm{in}}) = \min_c \frac{e^{\bar{s}_c(\mathbf{x})/\bar{\tau}}}{\sum_{j=1}^C e^{\bar{s}_j(\mathbf{x})/\bar{\tau}}}, \tag{13}$$

where $\tau$ and $\bar{\tau}$ are the temperature factors. We set $\tau$ and $\bar{\tau}$ to be $1$ and $5$ respectively, as we use different temperature factors when training.

Finally, the score function for OOD detection is the difference between MCM and MinCM:

$$S(\mathbf{x}) = S_{\mathrm{MCM}}(\mathbf{x}; \mathcal{Y}_{\mathrm{in}}) - S_{\mathrm{MinCM}}(\mathbf{x}; \mathcal{Y}_{\mathrm{in}}). \tag{14}$$

## E    COMPARISON WITH DUALCOOP.

A similar concept to LSN appears in DualCoOp (Sun et al., 2022a). DualCoOp focuses on multi-label recognition (MLR), where each input image includes multiple objects, and the model needs to identify the classes of all the objects correctly. Similar to our work, DualCoOp learns a pair of prompts to provide positive and negative contexts for the target class. Instead of using hand-crafted thresholding to determine positive labels, the learned prompts naturally result in a positive and a negative classifier, so the existence of the target class in the image can be easily decided by comparing their scores. However, although both DualCoOp and LSN propose negative prompts, their roles are different. In DualCoOp, negative prompts act as a threshold for positive prompts of the corresponding category, whereas in LSN, negative prompts focus on learning generic negative features across all other categories except the corresponding category. In DualCoOp, positive and negative prompts are optimized simultaneously with Asymmetric Loss, which makes the learning of negative logits intractable, resulting in DualCoOp tending to generate false negative responses, which is consistent with the observation in (Hu et al., 2023). In contrast, in LSN, the learning of positive and negative prompts is separated, and the learned negative prompts successfully focus on different regions than the positive prompts, as demonstrated in Figure 2. Besides, DualCoOp mainly focuses on multi-label recognition (MLR), which is a closed-set problem. This needs the class names of the unseen classes for testing. In contrast, LSN focuses on OOD detection, which is an open-set problem. This does not need unseen classes. We compare OOD detection performance of DualCoOp and LSN on ImageNet-100 with CLIP ResNet-50. As shown in Table 7, the result shows that LSN outperforms DualCoOp.

## F    COMPARISON WITH OTHER POS-HOC METHODS.

To enrich our analysis, we further conduct comparative experiments with some recent pos-hoc methods (Park et al., 2023; Djurisic et al., 2023; Sun et al., 2022b; Ahn et al., 2023). Note that here we simply replace the backbone used in these works with the CLIP ViT-b/16 model without further fine-tuning. The results on ImageNet-1k is shown in Table 8.

Our experimental results show that these post-hoc methods don't perform well on the raw CLIP model. We believe a major reason for this is the difference in training data. Many pos-hoc methods are designed on ImageNet pre-trained networks, which corresponds to the fact that only ID data are used during training. In contrast, when training CLIP, both ID data and OOD data are used. The difference in training data leads to different activations of OOD data.

Table 8: OOD detection performance on ImageNet-1k as ID.

| Methods | OOD Dataset | | | | | | | | | |
| | iNaturalist | | SUN | | Places | | Textures | | Average | |
| | FPR95↓ | AUROC↑ | FPR95↓ | AUROC↑ | FPR95↓ | AUROC↑ | FPR95↓ | AUROC↑ | FPR95↓ | AUROC↑ |
|---|---|---|---|---|---|---|---|---|---|---|
| MCM | 32.08 | 94.41 | 39.21 | 92.28 | 44.88 | 89.83 | 58.05 | 85.96 | 43.55 | 90.62 |
| NNGuide | 97.83 | 78.32 | 93.35 | 81.36 | 86.93 | 82.48 | 93.98 | 78.59 | 93.02 | 80.18 |
| ASH | 58.48 | 87.61 | 56.35 | 86.06 | 58.77 | 84.34 | 86.87 | 61.41 | 65.11 | 79.85 |
| KNN | 99.36 | 68.34 | 97.06 | 68.05 | 93.40 | 72.01 | 98.38 | 68.67 | 97.05 | 69.27 |
| LINe | 73.00 | 87.34 | 58.83 | 89.27 | 58.92 | 86.68 | 60.33 | 86.01 | 62.77 | 87.32 |
| LSN | 21.56 | 95.83 | 26.32 | 94.35 | 34.48 | 91.25 | 38.54 | 90.42 | 30.22 | 92.96 |

Table 9: On the effect of model architectures.

| Methods | OOD Dataset | | | | | | | | | |
| | iNaturalist | | SUN | | Places | | Textures | | Average | |
| | FPR95↓ | AUROC↑ | FPR95↓ | AUROC↑ | FPR95↓ | AUROC↑ | FPR95↓ | AUROC↑ | FPR95↓ | AUROC↑ |
|---|---|---|---|---|---|---|---|---|---|---|
| MCM/RN50x4 | 14.56 | 97.28 | 36.09 | 94.64 | 36.17 | 93.99 | 45.05 | 92.68 | 32.97 | 94.65 |
| LSN/RN50x4 | **5.37** | **98.64** | **14.36** | **96.84** | **18.47** | **96.03** | **11.78** | **97.73** | **12.49** | **97.31** |
| MCM/CLIP-B/32 | 10.01 | 97.80 | 36.19 | 94.33 | 35.67 | 94.11 | 40.78 | 92.96 | 30.66 | 94.80 |
| LSN/CLIP-B/32 | **3.87** | **99.11** | **10.69** | **97.23** | **15.37** | **96.85** | **9.43** | **98.24** | **9.84** | **97.86** |
| MCM/CLIP-B/16 | 18.13 | 96.77 | 36.45 | 94.54 | 34.52 | 94.36 | 41.22 | 92.25 | 32.58 | 94.48 |
| LSN/CLIP-B/16 | **4.93** | **98.92** | **8.23** | **97.98** | **12.82** | **97.19** | **8.26** | **98.11** | **8.56** | **98.05** |

## G   LSN WITH DIFFERENT NETWORK ARCHITECTURES.

To show the effectiveness of our method on different model architectures, we conducted experiments with CLIP-B/16, CLIP-B/32, and CLIP-RN50x4 models, respectively. As shown in Table 9, with different model structures, our method goes well beyond the baseline MCM.

## H   LSN WITH OTHER VLMS.

We further conduct experiments on other VLMs, such as BLIP (Li et al., 2022). As shown in Table 10, LSN also works well on BLIP.

## I   LSN UNDER DIFFERENT NUMBER OF ID CLASSES.

By varying numbers of ID classes, we performed experiments on ImageNet-10, ImageNet-20, ImageNet-100, and ImageNet-1k. The results are as shown in Table 11, indicating that LSN is robust to the number of ID classes.

## J   THE EFFECT OF THE NUMBER OF LABELED TRAINING SAMPLES PER CLASS.

To study the effect of different numbers of training set samples on the learned positive prompts and negative prompts, we vary the number of training set samples per class. As shown in Figure 7, the performance of both positive prompts and negative prompts increases gradually as the number of training samples increases, and the increase of negative prompts is higher than that of positive prompts. Since negative prompts do not perform well with small training sample sizes, we use a relatively large number of training samples in our experiments.

## K   MORE EXAMPLES ABOUT CLIP'S ABILITY ON UNDERSTANDING "NOT"

We exhibit more examples to show the CLIP's ability to understand "not"; see Figure 8. The results of the experiment are consistent with the conclusion in the formal paper. To test the model on more data, we use the OOD datasets iNaturalist, SUN, Places, and Texture on ImageNet benchmark to

Table 10: LSN with BLIP.

| Methods | OOD Dataset | | | | | | | | | |
| | iNaturalist | | SUN | | Places | | Textures | | Average | |
| | FPR95↓ | AUROC↑ | FPR95↓ | AUROC↑ | FPR95↓ | AUROC↑ | FPR95↓ | AUROC↑ | FPR95↓ | AUROC↑ |
|---|---|---|---|---|---|---|---|---|---|---|
| MCM | 46.45 | 93.37 | 78.48 | 81.76 | 77.14 | 81.05 | 48.10 | 90.92 | 62.54 | 86.77 |
| CoOp | 27.83 | 95.26 | 45.85 | 88.26 | 47.38 | 87.42 | 36.28 | 92.93 | 39.33 | 90.96 |
| CoOp + LSN | 16.33 | 96.73 | 39.48 | 90.94 | 41.76 | 90.42 | 31.59 | 94.27 | 32.29 | 93.09 |

Table 11: LSN under different number of ID classes.

| Methods | OOD Dataset | | | | | | | | | |
| | iNaturalist | | SUN | | Places | | Textures | | Average | |
| | FPR95↓ | AUROC↑ | FPR95↓ | AUROC↑ | FPR95↓ | AUROC↑ | FPR95↓ | AUROC↑ | FPR95↓ | AUROC↑ |
|---|---|---|---|---|---|---|---|---|---|---|
| ImageNet-10/MCM | 0.12 | 99.80 | 0.29 | 99.79 | 0.88 | 99.62 | 0.04 | 99.90 | 0.33 | 99.78 |
| ImageNet-10/LSN | 0.05 | 99.90 | 0.09 | 99.91 | 0.15 | 99.89 | 0.01 | 99.99 | 0.08 | 99.92 |
| ImageNet-20/MCM | 1.02 | 99.66 | 2.55 | 99.50 | 4.40 | 99.11 | 2.43 | 99.03 | 2.60 | 99.32 |
| ImageNet-20/LSN | 0.34 | 99.88 | 0.76 | 99.74 | 0.95 | 99.75 | 0.43 | 99.86 | 0.62 | 99.81 |
| ImageNet-100/MCM | 18.13 | 96.77 | 36.45 | 94.54 | 34.52 | 94.36 | 41.22 | 92.25 | 32.58 | 94.48 |
| ImageNet-100/LSN | 4.93 | 98.92 | 8.23 | 97.98 | 12.82 | 97.19 | 8.26 | 98.11 | 8.56 | 98.05 |
| ImageNet-1k/MCM | 32.08 | 94.41 | 39.21 | 92.28 | 44.88 | 89.83 | 58.05 | 85.96 | 43.55 | 90.62 |
| ImageNet-1k/LSN | 21.56 | 95.83 | 26.32 | 94.35 | 34.48 | 91.25 | 38.54 | 90.42 | 30.22 | 92.96 |

evaluate CLIP's ability to understand the meaning of "not." For every sample, we ask CLIP to pick between "a photo of a [class]" and "not a photo of a [class]," where "[class]" is the class name of ImageNet-1k. For each class, we count the number of samples categorized as "a photo of a [class]" and the number of samples categorized as "not a photo of a [class]." As shown in Figure 9 10 11 12, it is clear that CLIP tends to choose "not a photo of a [class]" as the correct description rather than "a photo of [class]".

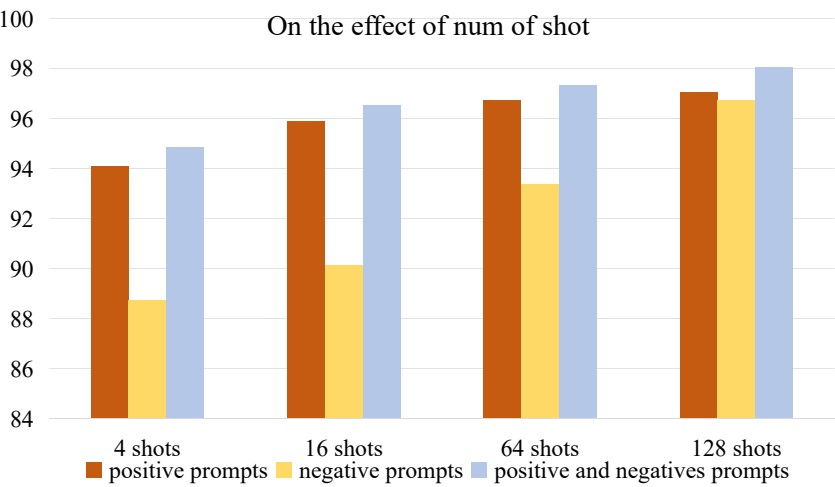

Figure 7: The effect on the number of labeled training samples per class.

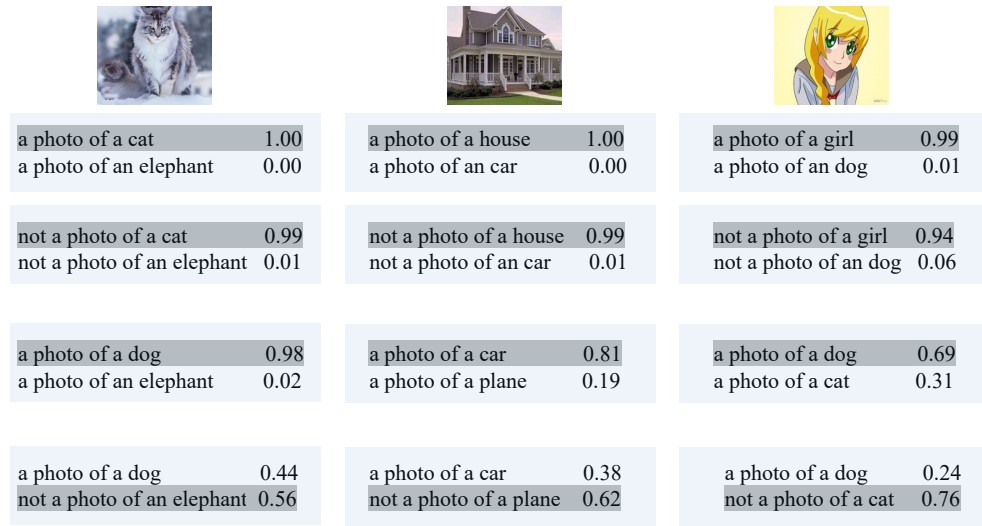

Figure 8: We explore the ability of the model to understand the meaning of "not." When the correct class name of a sample is given, CLIP tends to choose the one that has the correct class name as the sample's description, no matter whether "not" exists or not. But when the correct class name isn't given, CLIP tends to choose the one that has "not" as the sample's correct description.

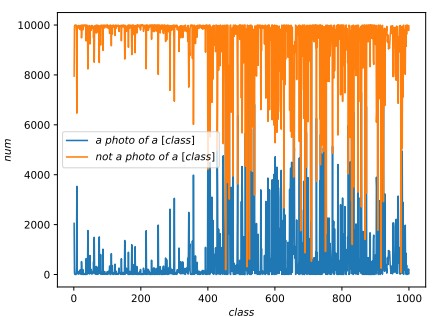

Figure 9: Statistical result on iNaturalist.

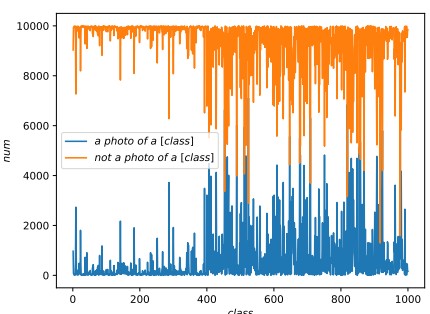

Figure 10: Statistical result on Places.

Figure 11: Statistical result on SUN.

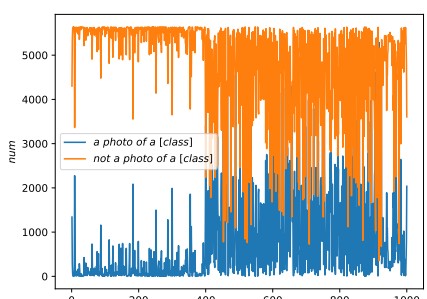

Figure 12: Statistical result on Texture.

