# OpenReview forum: "Out-of-Distribution Detection with Negative Prompts"
_ICLR.cc/2024/Conference — ICLR 2024 poster_

### Official Review · Reviewer_ayMs · 2023-10-22

**Soundness:** 2 fair
**Presentation:** 2 fair
**Contribution:** 2 fair
**Rating:** 5
**Confidence:** 4

**Summary:**

This paper focuses on the out-of-distribution detection problem, which aims to precisely classify samples of known categories, and accurately discern samples of unknown categroies. To facilitate the recognition the out-of-distribution examples, a CLIP-based method is proposed, where a set of learnable negative prompts for each class are introduced. Promising results are obtained compared to existing out-of-distribution detection methods.

**Strengths:**

1. This paper is clearly written and easy to follow. The weakness of the hand-crafted prompts is clearly interpreted and the motivation is reasonable.
2. The proposed approach is simple and intuitive.
3. Promising experimental results are achieved compared to existing OOD detection and prompt-based methods.

**Weaknesses:**

1. Meta-Net in Figure 6 is not introduced in the paper.
2. The motivation and model design of this paper are similar to DualCoOp. The authors claim that the proposed method could learn negative prompts to capture negative features compared to DualCoOp, but there is no visual or quantitative evidence to verify this statement. Besides, the OOD detection performance of the DualCoOp design is not experimentally verified.
3. Some existing OOD detection methods proposed in 2022 and 2023 are not compared or discussed[2][3][4].
4. It seems that the authors have submitted this paper with an ICLR 2023 template.

[1] Dualcoop: Fast adaptation to multi-label recognition with limited annotations.

[2] Out-of-Distribution Detection with Deep Nearest Neighbors.

[3] LINe: Out-of-Distribution Detection by Leveraging Important Neurons.

[4] Decoupling MaxLogit for Out-of-Distribution Detection

**Questions:**

In equation(9), the positive score and the negative score are obtained independently. If the obtained positive category is different from the negative category, could this lead to mistakenly recognizing a correct positive prediction as out-of-distribution?

---

> ### Author Response · Authors · 2023-11-17
>
> We sincerely thank you for your constructive comments! Please find our responses below.
>
> >**Q1:** Meta-Net in Figure 6 is not introduced in the paper.
>
> **A1:** Thanks for pointing out the problem. Accoridngly, we have added the following explanations in our revision.
>
> The Meta-Net in Figure 6 is a lightweight neural network built with a two-layer bottleneck structure (Linear-ReLU-Linear), with the hidden layer reducing the input dimension by 16×.
>
> >**Q2:** The motivation and model design of this paper are similar to DualCoOp.
>
> **A2:** Thank for your insightful comments and kind suggestion. Accordingly, we have added the following explanations to clarify the difference bewteen our method and DualCoOp.
>
> - The  roles of negative prompts in DualCoOp and LSN are different. In DualCoOp, negative prompts act as a threshold for positive prompts of the corresponding category, whereas in LSN, negative prompts focus on learning generic negative features across all other categories except the corresponding category.
> - In DualCoOp, positive and negative prompts are optimized simultaneously with Asymmetric Loss, which make the learning of negative logits intractable, resulting DualCoOp tend to generate false negative responses, which is consistent with the observation in DualCoOp++[1]. In contrast, in LSN, the learning of positive and negative prompts is separated, and the learned negative prompts successfully focus on different region than the positive prompts, as demonstrated in Figure2 in our paper.
> - DualCoOp mainly focuses on the **multi-label recognition (MLR)**, which is a **closed-set** problem. This needs class names of the unseen classes for testing. In contrast, LSN focuses on the **OOD detection**, which is an **open-set** problem. This does not need unseen classes.
>
> In response to your constructive comments, we conducted the following experiments on ImageNet-100 to hilight the difference from an empirical perspective. The corresponding results and discussions have been added to the revised paper. These results show that LSN can outperform DualCoOp.
> |          | Average |       |
> |----------|---------|-------|
> | Method   |  PFR95   | AUROC |
> | DualCoOp |  23.47   | 95.64 |
> | LSN      |  19.70   | 96.22 |
>
> >**Q3:** Some existing OOD detection methods proposed in 2022 and 2023 are not compared or discussed.
>
> **A3:** Thanks for your valuable comments. Following your kind suggrestion, we conducted comparative experiments with KNN and LINe on raw CLIP model. The results on ImageNet-1k are as follows:
>
> |                     |  Average |       |
> |---------------------|---------|-------|
> | Method              |  PFR95   | AUROC |
> | KNN(k=1)            |  97.05   | 69.27 |
> | KNN(k=10)           |  97.61   | 68.38 |
> | KNN(k=100)          |  98.52   | 61.80 |
> | LINe($\delta = 0$)    | 87.31   | 74.38 |
> | LINe($\delta = 0.04$) | 70.46   | 85.63 |
> | LINe($\delta = 0.07$) |  62.77   | 87.32 |
> | LSN                 |  30.22   | 92.96 |
>
> Our experimental results show that LSN outperforms these post-hoc methods on raw CLIP model. And the reason is the same as that in **A1** to **Reviewer Dyvx**, please see that.
>
> As the mentioned DML requires re-training the model, it is challenging to report detailed results. Thus, we will add the results when finishing our experiments. We have added these discussions and outstanding works in our revision.
>
> >**Q4:** It seems that the authors have submitted this paper with an ICLR 2023 template.
>
> **A4:** Thank you very much for the kind reminder, in our resubmitted version we have fixed this error.
>
> >**Q5:** In equation(9), the positive score and the negative score are obtained independently. If the obtained positive category is different from the negative category, could this lead to mistakenly recognizing a correct positive prediction as out-of-distribution?
>
> **A5:** We appreciate your careful review and indepth question. What you describe is exactly the reason we get positive score and negative score respectively. It can indeed happen that positive and negative categories are different, especially when the number of class is large. In this case, for ID samples:
>
> $$\max _c Softmax( \cos \left(\boldsymbol{f}(\mathbf{x}), \boldsymbol{g}\circ \boldsymbol{V}(l_c) \right)  / \tau) - \min _c Softmax( \cos \left(\boldsymbol{f}(\mathbf{x}), \boldsymbol{g}\circ \bar{\boldsymbol{V}}(l_c) \right) / \tau) > \max _c (Softmax( \cos \left(\boldsymbol{f}(\mathbf{x}), \boldsymbol{g}\circ \boldsymbol{V}(l_c) \right)  / \tau) -  Softmax( \cos \left(\boldsymbol{f}(\mathbf{x}), \boldsymbol{g}\circ \bar{\boldsymbol{V}}(l_c) \right) / \tau))$$
> Getting positive and negative scores independently can alleviate the problem you describe and this can be seen as an integration of positive classifier and negative classifier.
>
> Reference:
>
> [1]: DualCoOp++: Fast and Effective Adaptation to Multi-Label Recognition with Limited Annotations. Hu et al.

---

> ### Author Response · Authors · 2023-11-20
> **Welcome for more discussions**
>
> Dear Reviewer #ayMs,
>
> Thanks very much for your time and valuable comments.
>
> Here is a **summary of our response** for your convenience:
> - (1) **Missing information issues:** We have added a description of Meta-Net in our revision.
> - (2) **Novelty issues:** DualCoOp differs from LSN in the following three points:
>
>  **Negative prompts of DualCoOp and LSN serve different purposes;**
>
>  **Negative prompts of DualCoOp and LSN are learned in different ways;**
>
>  **DualCoOp and LSN focus on different tasks.**
>
>  And in OOD detection task, LSN outperforms DualCoOp.
> - (3) **Related work issues:** Following your constructive comments, we have discussed related works including KNN, LINe and DML.
> - (4) **Score function issues:** Inconsistencies between positive category and negative category can indeed occur. When it happens, the influence on ID data scores can be mitigated by using our scoring function.
>
> We understand you're busy. But as the window for responding and paper revision is closing, would you mind checking our response and confirm whether you have any further questions? We are very glad to provide answers and revision to your further questions.
>
> Best regards and thanks,
>
> Authors of #1826

---

> ### Author Response · Authors · 2023-11-21
> **Window for discussion and revision is closing**
>
> Dear Reviewer #ayMs,
>
> Thanks a lot for your time in reviewing and insightful comments, according to which we have carefully revised the paper to answer the questions. We sincerely understand you’re busy. But since the discussion due is approaching, would you mind checking the response and revision to confirm where you have any further questions?
>
> We are looking forward to your reply and happy to answer your further questions.
>
> Best regards
>
> Authors of #1826

---

> ### Author Response · Authors · 2023-11-22
> **Window for discussion and revision is closing**
>
> Dear Reviewer #ayMs,
>
> Thanks a lot for your time in reviewing and insightful comments. We sincerely understand you're busy. Would you mind checking the response and revision to see if you have any further questions?
>
> We are looking forward to your reply.
>
> Best regards,
>
> Authors of #1826

---

> ### Author Response · Authors · 2023-11-22
> **Window for discussion and revision is closing**
>
> Dear Reviewer #ayMs,
>
> Thanks a lot for your time in reviewing and insightful comments. We sincerely understand you're busy. Would you mind checking the response and revision to see if you have any further questions?
>
> We are looking forward to your reply.
>
> Best regards,
>
> Authors of #1826

---

### Official Review · Reviewer_ZZhi · 2023-10-30

**Soundness:** 3 good
**Presentation:** 3 good
**Contribution:** 3 good
**Rating:** 6
**Confidence:** 5

**Summary:**

In this paper, the authors present a negative prompt tuning method with CLIP to improve the OOD performance. Specifically, the authors learn class-specific prompts for each category. A semantic orthogonality loss is also applied to encourage diverse negative prompts. The negative prompts are also considered in the evaluation.

**Strengths:**

1. The proposed LSN method achieves good performance over baselines on various OOD benchmarks.
2. The authors provide sufficient ablation studies to show the effectiveness of each proposed component.
3. The proposed idea is simple and easy to understand.

**Weaknesses:**

1. This paper is related to negative learning or learning with complementary labels. The authors may consider adding some related discussion in the related work section.
2. The proposed method may double the training and inference time with the negative prompts.
3. I found a related work that the authors may add discussion in the related work section:
" How Does Fine-Tuning Impact Out-of-Distribution Detection for Vision-Language Models? , IJCV 2023."

**Questions:**

1. What's the ID dataset in Table 3?
2. Why do CoOp/CoCoOp and CoOp/CoCoOp + LSN achieve the same ID results in Table 1 and Table 2?
3. The ID results of CoOp/CoCoOp appear to be significantly lower than other baselines such as NPOS in Table 2. Can the authors explain some reasons?

Some minor suggestions that do not affect my final rating:
1. It is suggested to use `\citep' rather than `\cite' in the latex code
2. Typo: 'we use we use' at the bottom of page 6

---

> ### Author Response · Authors · 2023-11-17
>
> We sincerely thank you for your constructive comments! Please find our responses below.
>
> >**Q1:** This paper is related to negative learning or learning with complementary labels. The authors may consider adding some related discussion in the related work section.
>
> **A1:**  Thanks for your constructive suggestion. We have added a section of **Complementary Label Learning** to the related work in our revised version
>
> >**Q2:** The proposed method may double the training and inference time with the negative prompts.
>
> **A2:** We agree with your point. Due to the use of both positive and negative prompts, we indeed double our training and inference time. We will add the corresponding overhead analysis into our revision.
>
> >**Q3:** I found a related work that the authors may add discussion in the related work section: " How Does Fine-Tuning Impact Out-of-Distribution Detection for Vision-Language Models? , IJCV 2023."
>
> **A3:** Thanks for mentioning the outstanding paper. This paper investigates the effect of fine-tuning on OOD detection in large vision-language models and the maximum concept matching (MCM) score is highlighted as effective. We have added the above discussion to the related work.
>
> >**Q4:** What's the ID dataset in Table 3?
>
> **A4:**  We apologize for the missing information. Accordingly, we have added the following explanations into our revised paper.
>
> For the small-scale dataset, we follow the setting in ZOC[1]. For CIFAR10, 6 classes are selected as ID classes, and the 4 remaining classes are selected as OOD classes. Experiments are conducted on 5 different splits. For CIFAR+10, 4 non-animal classes from CIFAR10 are selected as ID classes, and 10 animal classes from CIFAR100 are selected as OOD classes. For CIFAR+50, 4 non-animal classes from CIFAR10 are selected as ID classes, and all 50 animal classes from CIFAR100 are selected as OOD classes. For CIFAR100, 20 consecutive classes are selected as ID classes. The remaining 80 classes are selected as OOD classes. For TinyImagenet, 20 classes are used as ID classes, and the remaining 180 classes are used as OOD classes. All the split is the same as ZOC. We give detailed split information in Table6.
>
>
> >**Q5:** Why do CoOp/CoCoOp and CoOp/CoCoOp + LSN achieve the same ID results in Table 1 and Table 2?
>
> **A5:** Thanks for pointing out this potentially confusing description.
>
> In our experiments we notice that the classification performance of negative prompts is slightly less than that of positive prompts. We are not committed to improving ID classification performance. To avoid further trouble, we only use positive prompts when determining the classes of ID samples. We have clarified this point in our revision.
>
> Thanks again for your careful review.
>
> >**Q6:** The ID results of CoOp/CoCoOp appear to be significantly lower than other baselines such as NPOS in Table 2. Can the authors explain some reasons?
>
> **A6:** We do not run NPOS on ImageNet-1k because it will consume a lot of resources. We get their experimental results directly from the original NPOS paper. We believe that the gap in ID classification performance comes from the difference in the amount of training data as well as the difference in training methods. NPOS uses the full ImgaNet-1k training set to fine-tune the CLIP model, which allows the fine-tuned model to fit better to the ImgaNet-1k test dataset. In contrast, LSN uses only 64 samples per class on ImageNet-1k and only learns prompts  without making changes to the model.
>
> >**Q7:** 1: It is suggested to use ``\citep`` rather than ``\cite`` in the latex code; 2: Typo: ``we use we use`` at the bottom of page 6.
>
> **A7:** Thank you very much for the kind reminder; in our resubmitted version, we have fixed these errors.

---

> ### Author Response · Authors · 2023-11-20
> **Welcome for more discussions**
>
> Dear Reviewer #ZZhi,
>
> Thanks very much for your time and valuable comments.
>
> Here is a **summary of our response** for your convenience:
>
> - (1) **Related work issues:** In response to your constructive suggestions, we have added a discussion of the work mentioned in your comments and added a section on **Complementary label learning** to the related work.
> - (2) **Training and testing time issues:** We have added a section of **Limitations** in the Appendix and discussed this issue.
> - (3) **Missing information issues:** We describe Table3 in detail and give specific split information in the Appendix.
> - (4) **ID ACC issues:** When we determine the class of ID samples, we only use positive prompts, resulting in the same ID results of CoOp/CoCoOp and CoOp/CoCoOp+LSN. And the reason NPOS achieves high ID ACC is that it uses the full ImageNet dataset to fine-tune CLIP.
>
> We understand you're busy. But as the window for responding and paper revision is closing, would you mind checking our response and confirm whether you have any further questions? We are very glad to provide answers and revision to your further questions.
>
> Best regards and thanks,
>
> Authors of #1826

---

> ### Author Response · Authors · 2023-11-21
> **Window for discussion and revision is closing**
>
> Dear Reviewer #ZZhi,
>
> Thanks a lot for your time in reviewing and insightful comments, according to which we have carefully revised the paper to answer the questions. We sincerely understand you’re busy. But since the discussion due is approaching, would you mind checking the response and revision to confirm where you have any further questions?
>
> We are looking forward to your reply and happy to answer your further questions.
>
> Best regards
>
> Authors of #1826

---

### Official Review · Reviewer_r91y · 2023-10-31

**Soundness:** 3 good
**Presentation:** 4 excellent
**Contribution:** 3 good
**Rating:** 8
**Confidence:** 4

**Summary:**

In this paper, the authors perform OOD detection based on generic features learned from a large pre-trained language-vision model by matching the similarity between image features and features of learned positive prompts and negative prompts.
The core innovation of this paper is the proposed LSN module to learn a set of negative prompts for each ID category to help the network to comprehend the concept of "not." They mine general negative features that are not present in a category but are present in all other categories by proposing a new loss in prompt learning. In the test, the MCM scores of the cosine similarity of positive prompts and negative prompts to image features are used as OOD detection metrics. Extensive experiments on various ood detection benchmarks have been conducted to demonstrate the effectiveness of the method proposed in this work.

**Strengths:**

1、This work utilizes the generic feature extraction capability of CLIP and does not need to finetune the image encoder and text encoder. It only needs to learn the appropriate positive and negative prompts by LSN for OOD detection. Therefore this method has high generality and low complexity.
2、SOTA performance is achieved in different benchmark experiments.

**Weaknesses:**

The overall prompt learning approach is still based on CoOp without a lot of innovation.

**Questions:**

1, This method is very dependent on the features learned by CLIP. If the features extracted by CLIP itself for some categories of images are not strongly discriminative, the effect of learning the prompts based on these features may be poor.
2, The way to learn negative prompts is to mine the general negative features that each class of samples does not have but all other classes have, i.e., the negative classifier produces low activation values for that class and high activation values for other classes. Does this result in learning what are actually generic background features rather than general features that all other classes have?

---

> ### Author Response · Authors · 2023-11-17
>
> We sincerely thank you for your constructive comments! Please find our responses below.
>
> >**Q1:** The overall prompt learning approach is still based on CoOp without a lot of innovation.
>
> **A1:** Thank you for your valuable comment. We agree with your point that LSN is built upon the success of CoOp, while we would like to provide more explanations to highlight the novel design for OOD detection.
>
> Although our approach to learn negative prompts looks basically the same as CoOp,  there is a fundamental difference between positive prompt learning and negative prompt learning. In positive prompt learning, using a shared prompt across classes is enough to achieve good performance. This is because in positive prompt learning, the positive features of each class are carried by the class names, and the positive prompt is only used to calibrate these features to the downstream dataset. In negative prompt learning, on the other hand, the situation is completely different. The negative features need to be contained in the learned negative prompts, and the class names appear to be less important. Diversity is the key to the success of negative prompts. We illustrate this in Table 5.
>
> >**Q2:** If the features extracted by CLIP itself for some categories of images are not strongly discriminative, the effect of learning the prompts based on these features may be poor.
>
> **A2:** We wholeheartedly agree with your observation and acknowledge that this situation is indeed inevitable in the current framework. Inspired by your insightful comments, we will explore the promising direction, like finetuning a "degenerated" CLIP model for promoting OOD detection.
>
> >**Q3:** The way to learn negative prompts is to mine the general negative features that each class of samples does not have but all other classes have, i.e., the negative classifier produces low activation values for that class and high activation values for other classes. Does this result in learning what are actually generic background features rather than general features that all other classes have?
>
> **A3:** We agree with your indepth point. It is indeed possible for the model to learn generic background features as negative features. However, it is noteworthy that even when background features are learned, they can at times serve as effective negative features. For example, consider the ID classes Horse, Wolf, and Pet Cat, and the OOD class Tiger. Due to similar positive features with pet cats, tigers can easily be mistaken for ID samples. But if we learn further about negative features the situation is different. Since horses and wolves are often found in grassland and pet cats are unlikely to be found in grassland, then the background grassland is likely to be learned as a negative feature for pet cats. The background of tigers is also usually grassland, and then the classifier can easily determine that tigers are not pet cats.

---

> ### Author Response · Authors · 2023-11-20
> **Welcome for more discussions**
>
> Dear Reviewer #r91y,
>
> Thanks very much for your time and valuable comments.
>
> Here is a **summary of our response** for your convenience:
>
> - (1) **Novelty issues:** Positive prompt learning is fundamentally different from negative prompt learning. Negative features need to be included in the learned negative prompts. Class names do not provide valid negative features.
> - (2) **Validity issues:** It does happen that the learned negative features are background features. However, background features are also useful for OOD detection.
>
> We understand you're busy. But as the window for responding and paper revision is closing, would you mind checking our response and confirm whether you have any further questions? We are very glad to provide answers and revision to your further questions.
>
> Best regards and thanks,
>
> Authors of #1826

---

> ### Author Response · Authors · 2023-11-21
> **Window for discussion and revision is closing**
>
> Dear Reviewer #r91y,
>
> Thanks a lot for your time in reviewing and insightful comments, according to which we have carefully revised the paper to answer the questions. We sincerely understand you’re busy. But since the discussion due is approaching, would you mind checking the response and revision to confirm where you have any further questions?
>
> We are looking forward to your reply and happy to answer your further questions.
>
> Best regards
>
> Authors of #1826

---

### Official Review · Reviewer_Dyvx · 2023-11-01

**Soundness:** 2 fair
**Presentation:** 3 good
**Contribution:** 2 fair
**Rating:** 6
**Confidence:** 5

**Summary:**

The paper proposes how to use CLIP for OOD detection with negated prompts that include the negation word 'not'. Using the learnable prompts embeddings, the method trains the model by freezing the CLIP encoders based on a contrastive loss. The proposed method shows improvement over the baseline CLIP for OOD detection.

**Strengths:**

1. The paper is clearly written with descriptive visual figures.
2. Experiments have been extensively conducted across a set of diverse datasets including small ones and large ones.

**Weaknesses:**

1. Comparison and related works to state-of-the-arts are missing (e.g., NNGuide [1], ASH [2], CLIPN [3], [4])
2. The performance is too behind the state-of-the-art
3. The main concept of the paper is too similar to CLIPN
4. The performance improvement is very marginal

[1]
[2]
[3] Wang, Hualiang, et al. "Clipn for zero-shot ood detection: Teaching clip to say no." Proceedings of the IEEE/CVF International Conference on Computer Vision. 2023.
[4] Ming, Yifei, et al. "Delving into out-of-distribution detection with vision-language representations." Advances in Neural Information Processing Systems 35 (2022): 35087-35102.

**Questions:**

I suggest the authors to properly address the above weaknesses.

---

> ### Author Response · Authors · 2023-11-17
>
> We sincerely thank you for your constructive comments! Please find our responses below.
>
> >**Q1:** Comparison and related works to state-of-the-arts are missing. (e.g., NNGuide, ASH, CLIPN)
>
> **A1:** Thanks for your contructive comments. In accordance with your suggestion, we conduct comparison experiments with NNGuide and ASH to enrich our analysis. The results on ImageNet-1k are as follows:
> |                                | Average |       |
> |--------------------------------|---------|-------|
> | Method                         | PFR95   | AUROC |
> | NNGuide(k=1)                   | 93.02   | 80.18 |
> | NNGuide(k=10)                  | 94.46   | 78.96 |
> | NNGuide(k=100)                 | 96.38   | 75.52 |
> | ASH-B(@90) |  91.76   | 66.52 |
> | ASH-B(@65) | 85.05   | 68.17 |
> | ASH-P(@90) | 83.10  | 69.78 |
> | ASH-P(@65) | 65.11  | 79.85 |
> | ASH-S(@90) | 78.10  | 73.20 |
> | ASH-S(@65) | 65.24  | 79.25 |
> | LSN        | 30.22   | 92.96 |
>
> Our experimental results show that LSN outperforms these post-hoc methods on raw CLIP model. We believe the reason for the poor performance of these pos-hoc methods is that the **the difference in training data**. Many pos-hoc methods are designed on ImageNet pre-trained networks, where **only ID datas are used during training**. In contrast, when training CLIP, **both ID datas and OOD datas are used**. The difference in training data leads to different activation of OOD data.  Another reason is that the pos-hoc method relies heavily on the choice of hyperparameters. The hyperparameters need to be re-selected on different models.
>
> CLIPN also employes CLIP to perform OOD detection, coming up with ideas similar to ours. We compare the best results of CLIPN and LSN on ImageNet-1k with CLIP ViT-B-16 as follows:
>
> |        | iNaturalist |           | SUN       |           | Places    |           | Textures  |           | Average   |           |
> |--------|-------------|-----------|-----------|-----------|-----------|-----------|-----------|-----------|-----------|-----------|
> | Method | PFR95       | AUROC     | PFR95     | AUROC     | PFR95     | AUROC     | PFR95     | AUROC     | PFR95     | AUROC     |
> | CLIPN  | 23.94       | 95.27     | **26.17** | 93.93     | **33.45** | **92.28** | 40.83     | **90.93** | 31.10     | **93.10** |
> | LSN    | **21.56**   | **95.83** | 26.32     | **94.35** | 34.48     | 91.25     | **38.54** | 90.42     | **30.22** | 92.96     |
>
> - Experimental results show that LSN achieves comparable performance with CLIPN.
> - We have to point out that **this is not a fair comparison**. This is because, to endow OOD detector with the ability to say "no", CLIPN leverages a large-scale dataset [1] (containing 3.3 million image-text pairs) to finetune the text encoder.
> - These results highlight the contribution of the proposed approach to learn negative prompts. Specifically, LSN only needs to use few shot samples to learn negative prompts, which can be done in a very short time.
>
> Inspired by your constructive comments, we summarize the differences between LSN and CLIPN as follows.
>
> - LSN only requires learning negative prompts at a small computational cost without training "no" text encoder and achieves performance comparable with CLIPN.
> - LSN points out that the diversity of negative prompts is the key to success, and the use of class-specific negative prompts leads to a reduction in FPR95 from 81.27% to 21.94% compared to the use of class-shared negative prompts, please see Table 5.
> - LSN focuses on prompt learning and points out how negative prompt learning differs from positive prompt learning, while CLIPN focuses more on learning "no" text encoder.
>
> We will add the above results and discussion into our revision to highlight the difference bewteen our work and the mentioned work. Thanks again for mentioning these outstanding papers.
>
> >**Q2:** The performance is too behind the state-of-the-art.
>
> **A2:** Thanks for your valuable comments. Accordingly, we have added the following explanations to our revision.
>
> - We are focused on improving the OOD detection performance of pre-trained vision-language models such as CLIP. As can be seen in **A1**, our method can outperform NNguide and ASH, the recent SOTA pos-hoc methods.
> - Moreover, when compared to CLIPN, our method achieves comparable performance without the need for large-scale datasets and significantly computation overhead.
>
>
> >**Q3:** The performance improvement is very marginal.
>
> **A3:** As mentioned above, we get the comparable performance at a much smaller cost compared to CLIPN. And compared to standard prompt learning CoOp and CoCoOp, LSN's FPR95 on ImageNet1k decreased by 6.86% and 8.41%, respectively.
>
> Reference:
> [1] Conceptual captions: A cleaned, hypernymed, image alt-text dataset for automatic image captioning. Sharma et al. ACL2018.

---

> > ### Comment · Reviewer_Dyvx · 2023-11-17
> > **Thank you for your responses**
> >
> > I sincerely appreciate the detailed responses from the authors. I am mostly satisfied with the responses, and now I understand at which points the proposed LSN is superior to CLIPN. But I am still not sure if the overall contribution is sufficient enough.
> >
> > The paper's main contribution is the demonstration that the prompt tuning works for OOD detection. But I am worried under the following the aspects:
> >
> > [Practical usefulness]:
> > 1. Although the method outperforms other OOD detectors using CLIP encoder on ImageNet-1k, the average performance of FPR95 30% seems not satisfactory; most of the recent methods achieve below 25% of FPR95 [5] simply using light-weight ResNet-50.
> > 2. The method is restricted to CLIP-type encoders. Hence it may be difficult to apply the method on edge device applications.
> >
> > [Analytical Profoundness]
> > 3. I appreciate the descriptive explanation given in Fig. 4; they are insightful to me. But the experiment scope is limited with respect to the encoder variation. Would this method also be effective to other CLIP-type encoders such as BLIP?
> >
> > [Novelty]
> > 4. The idea of fine-tuning is not really new. The method itself is not really impactful although I really appreciate that the method is well presented in an organized manner.
> > 5. During the review, I also found [6], which is also an extension of CoOp. The contribution of LSM seems to largely overlap with [6].
> >
> > Due to the above reasons, I hesitate to increase the score.
> >
> > [5] LHAct: Rectifying Extremely Low and High Activations for Out-of-Distribution Detection, ACMMM2023
> > [6] LoCoOp: Few-Shot Out-of-Distribution Detection via Prompt Learning, NeurIPS2023

---

> > > ### Author Response · Authors · 2023-11-18
> > >
> > > Thank you for your prompt reply and constructive comments! Please find our responses below.
> > >
> > > >**Q1:** The average performance of FPR95 30% seems not satisfactory; most of the recent methods achieve below 25% of FPR95 simply using light-weight ResNet-50.
> > >
> > > **A1:** Thank you for your valuable comment. These methods do achieve surprising performance. However, a huge advantage of LSN over them is generalizability. Thanks to the power of CLIP,  **LSN can be easily and quickly generalized to new OOD tasks**. When generalized to a new OOD task, LSN requires only a small number of ID samples and ID class names. On the contrary, for these methods you mentioned, for each new OOD detection task, one would have to collect enough ID data and spend a lot of time re-training a model. This is not even feasible in cases where data for certain classes are difficult to collect. In addition, these methods require re-selection of hyperparameters under different models or different number of ID classes. This is also a major disadvantage of them.
> > >
> > > >**Q2:** The method is restricted to CLIP-type encoders. Hence it may be difficult to apply the method on edge device applications.
> > >
> > > **A2:** Thank you for your insightful comments! A straightforward idea is that one could simply download the output of CLIP text encoder instead of entire CLIP text encoder to the edge device and use it as the classification head for CLIP image encoder, so that CLIP behaves like a normal single-modal model on the edge device. When it is time to generalize to a new task, we simply re-download the new output of CLIP text encoder.
> > >
> > > >**Q3:** The experiment scope is limited with respect to the encoder variation. Would this method also be effective to other CLIP-type encoders such as BLIP?
> > >
> > > **A3:** We apologize for focusing more on the works of prompt learning and neglecting the implementation on the BLIP architecture. We further explore the model's ability to understand "not" on BLIP, MiniGPT-4, BLIP2 and GPT-4. For BLIP, given a picture of a *``dog``*, we asked how well it matches the text inputs *``"not a photo of a dog"``* and *``"not a photo of a cat"``*, and the results are 0.3335 and 0.1998, respectively, which indicates that BLIP is also **unable** to understand the meaning of "not". On MiniGPT-4 and BLIP2, given a picture of a dog, we input the question *``"Does this is not a photo of a dog or not a photo of a cat?"``*, MiniGPT4 gives the **wrong** answer *``"not a photo of a dog"``*, while BLIP2 gives the **correct** answer *``"not a photo of a cat"``*. As for GPT-4, it gives the correct answer, as expected. This shows that the problem of not being able to understand the meaning of "not" exists in many VLMs.
> > >
> > > We believe our approach also works on BLIP because its Image Text Matching module doesn't have much difference with CLIP. Due to time constraints, we can't provide experimental results at this time. We will do the corresponding experiments later, and add these discussion into our revision. Your comments make our work better, thank you!
> > >
> > > >**A4:** The idea of fine-tuning is not really new.
> > >
> > > **Q4:** Your concern is valid. But our work goes beyond fine-tuning; more importantly, we propose to mine negative features by learning negative prompts based on CLIP, and the effectiveness of negative features for OOD detection has been clearly illustrated in  **Negative classifier** section in our work.
> > >
> > > >**Q5:** During the review, I also found [6], which is also an extension of CoOp. The contribution of LSM seems to largely overlap with [6].
> > >
> > > **A5:** Thanks for mentioning this outstanding work! We believe that LSN are the opposite and complementary to the idea of LoCoOp. LoCoOp focuses on **learning positive prompts**, which is regularized by using the backgrounds information in the image as OOD information, such that only ID information is included in the learned positive text embeddings. In contrast, we focus on **negative prompt learning**, aiming to include negative features (negative features of a class can be considered as OOD information for that class) in negative text embeddings by learning negative prompts. Performance can be further improved when we use LoCoOp to learn positive prompts and LSN to learn negative prompts. We validate our idea on ImageNet-100. The experimental results are as follows:
> > > |            | iNaturalist |       | SUN   |       | Places |       | Textures |       | Average |       |
> > > |------------|-------------|-------|-------|-------|--------|-------|----------|-------|---------|-------|
> > > | Method     | FPR95       | AUROC | FPR95 | AUROC | FPR95  | AUROC | FPR95    | AUROC | FPR95   | AUROC |
> > > | LSN        | 5.74        | 98.32 | 12.42 | 97.53 | 14.62  | 96.89 | 9.17     | 97.82 | 10.49   | 97.64 |
> > > | LoCoOp     | 6.83        | 98.14 | 11.39 | 97.76 | 12.47  | 97.03 | 13.72    | 97.12 | 11.10   | 97.51 |
> > > | LSN+LoCoOp | 4.24        | 98.67 | 12.19 | 97.65 | 12.39  | 97.16 | 5.38     | 98.46 | 8.55    | 97.99 |

---

> > > ### Author Response · Authors · 2023-11-20
> > > **Welcome for more discussions**
> > >
> > > Dear Reviewer #Dyvx,
> > >
> > > Thanks very much for your time and valuable comments.
> > >
> > > Here is a **summary of our response** for your convenience:
> > >
> > > - (1) **Performance issues:** We are committed to improving the OOD detection performance of CLIP, which has a huge advantage of **generalizability**. And in this approach, we achieve SOTA performance.
> > > - (2) **Application issues:** Lower requirements for edge devices by storing only the output of the text encoder.
> > > - (3) **Model issues:** The inability of VLMs to understand the meaning of "not" is widespread. We believe that LSN is also effective on BLIP because BLIP is not fundamentally different from CLIP. We will give concrete experimental results in a few days.
> > > - (4) **Novelty issues:** We go beyond fine-tuning. More importantly, we propose the use of negative features for OOD detection and intuitively show its effectiveness.
> > > - (5) **Related work issues:** LoCoOp focuses on learning positive prompts and LSN focuses on learning negative prompts. The two are complementary and do not overlap. When combining the two, performance is further improved.
> > >
> > > We understand you're busy. But as the window for responding and paper revision is closing, would you mind checking our response and confirm whether you have any further questions? We are very glad to provide answers and revision to your further questions.
> > >
> > > Best regards and thanks,
> > >
> > > Authors of #1826

---

> > > ### Author Response · Authors · 2023-11-21
> > > **Window for discussion and revision is closing**
> > >
> > > Dear Reviewer #Dyvx,
> > >
> > > Thanks a lot for your time in reviewing and insightful comments, according to which we have carefully revised the paper to answer the questions. We sincerely understand you’re busy. But since the discussion due is approaching, would you mind checking the response and revision to confirm where you have any further questions?
> > >
> > > We are looking forward to your reply and happy to answer your further questions.
> > >
> > > Best regards
> > >
> > > Authors of #1826

---

> > > > ### Comment · Reviewer_Dyvx · 2023-11-21
> > > > **Thank you for the response**
> > > >
> > > > Thank you for the response. I truely appreciate the authors' prompt responses. But honestly, the concerns still remain.
> > > >
> > > > Q1. I understand CoOp variants outperform conventional OOD detectors in the few-shot setting. But this is obvious because CLIP has been pretrained on a very large-scale dataset. Is LSN then scalable such that if it is fine-tuned by full images of IN-1k, then can it achieve below FPR95 25%?
> > > >
> > > > Q3. The score of limited achirecture types is concerning. This may implicate the proposed method can work only for out-dated underperforming VLM such as CLIP. Would LSN be effective for BLIP2 as well?
> > > >
> > > > Q5. The reason behind the superiority of LSN over LoCoOp is not clear. LSN uses more parameters, and its mining mechanism adds extra cost. Is LSN better than LoCoOp really bacause LSN is focused on negative prompt learning? Why then is negative prompt learning better than positive prompt learning?
> > > >
> > > > I understand the conceptual orthogonality to LoCoOp. But would LSN always imporve LoCoOp (e.g., IN-1k)?
> > > >
> > > > ---
> > > > Overall, the contribution of the paper is there, but the advancement is marginal in terms of novelty and performance. In my perspective, the paper is a borderliner one.

---

> > > > > ### Author Response · Authors · 2023-11-21
> > > > >
> > > > > Thank you for your prompt responses! Please find our responses below.
> > > > >
> > > > > >Q1: I understand CoOp variants outperform conventional OOD detectors in the few-shot setting. But this is obvious because CLIP has been pretrained on a very large-scale dataset. Is LSN then scalable such that if it is fine-tuned by full images of IN-1k, then can it achieve below FPR95 25%?
> > > > >
> > > > > A1: Fine-tuning CLIP using the full ImageNet1k is a time-consuming job, and I am currently unsure how LSN performs on the fine-tuned CLIP. However, we believe that fine-tuning CLIP defeats our original purpose, and is not a wise choice. LSN can be viewed as a 'generalized' OOD detection method that can be quickly adapted to different OOD detection tasks. On the contrary, the method you mentioned is a specialized one. The model needs to be retrained and hyperparameters need to be reselected for different OOD detection tasks. It is not fair to compare a generic method with a specialized method.
> > > > >
> > > > > >Q3: The score of limited architecture types is concerning. This may implicate the proposed method can work only for out-dated underperforming VLM such as CLIP. Would LSN be effective for BLIP2 as well?
> > > > >
> > > > > A3: Our approach is based on prompt learning. And prompt learning is widely applicable on various models. There is no reason to think that our approach is limited to CLIP. And, for those more advanced models, such as BLIP2, when equipped with advanced language models, we find that it is capable of understanding the meaning of "not". In this case, instead of learning the negative prompts, we can directly use the hand-designed negative prompts "not a photo of a" to achieve a dissimilarity matching. For the time being, we have only tested BLIP2's ability to understand "not" on the APP provided by Hugging Face, and we will give details of the OOD detection performance of using both "a photo of a" and "not a photo of a" as prompts on BLIP2 in the future.
> > > > >
> > > > > >Q5: The reason behind the superiority of LSN over LoCoOp is not clear. LSN uses more parameters, and its mining mechanism adds extra cost. Is LSN better than LoCoOp really bacause LSN is focused on negative prompt learning? Why then is negative prompt learning better than positive prompt learning? I understand the conceptual orthogonality to LoCoOp. But would LSN always imporve LoCoOp (e.g., IN-1k)?
> > > > >
> > > > > A5: I am not certain whether LSN can outperform LoCoOp. In the ImageNet-100 experiment above, we simply set $K=50$ and $\lambda=0.25$ for LoCoOp since the authors don't conduct the corresponding experiments. Learning both positive and negative prompts does add extra cost. Moreover, learning negative prompts alone is not as effective as learning positive prompts alone. However, the learned negative prompts is always an effective complement to positive prompts and can always bring performance improvement to positive prompts. On the contrary, learning multiple positive prompts does not lead to performance improvement. These points are illustrated in Table 4 in our paper. On ImageNet-1k, LSN can also improve LoCoOp, with the following results:
> > > > >
> > > > > |              | iNaturalist |       | SUN   |       | Places |       | Textures |       | Average |       |
> > > > > |--------------|-------------|-------|-------|-------|--------|-------|----------|-------|---------|-------|
> > > > > | Method       | FPR95       | AUROC | FPR9595 | AUROC | FPR95  | AUROC | FPR95    | AUROC | FPR95   | AUROC |
> > > > > | LoCoOp       | 23.06       | 95.45 | 32.70 | 93.35 | 39.92  | 90.64 | 40.23    | 91.32 | 33.98   | 92.69 |
> > > > > | CoOp + LSN   | 23.48       | 95.47 | 29.84 | 93.45 | 36.43  | 90.87 | 38.16    | 89.52 | 31.97   | 92.33 |
> > > > > | LoCoOp + LSN | 18.56       | 96.03 | 27.39 | 93.72 | 34.88  | 91.24 | 37.25    | 92.17 | 29.52   | 93.29 |

---

> > > > > > ### Comment · Reviewer_Dyvx · 2023-11-21
> > > > > > **Thank you for the response**
> > > > > >
> > > > > > Thank you for the prompt response again.
> > > > > >
> > > > > > Q1. By 'fine-tuning' I mean fine-tuning of the learnable input prompt weights (i.e., 'prompt learning' by LSN). I assume that the set $S$ in the subscript of the expectation in Eqs. (4) and (5) are your training set for prompt learning. As far as I understand, based on your responses comments, this set is not the full ImageNet-1k training set. If you increase the size of $S$ to the full train fold of IN-1k, then would it correspondingly increase the performance, and achieve the SOTA?
> > > > > > (By the way, the notations are quite confusing. $S$ is elsewhere defined as the score function while in the Eqns (4) and (5) it seems not the case.)
> > > > > >
> > > > > > One major concern with CLIP-based OOD detection methods is that they only show few-shot results without showing results that utilize full training set. Hence, they lack to show scalability with respect to the train data size.
> > > > > >
> > > > > > Q3. Hopefully, the LSN works on BLIP2 as well; I am expecting the result.
> > > > > >
> > > > > > Q5. Thank you for the authors' reasonable and convincing points.
> > > > > >
> > > > > > Overall, my remaining concerns are only on Q1 and Q3.

---

> > > > > > > ### Author Response · Authors · 2023-11-22
> > > > > > > **Would you mind raising the score**
> > > > > > >
> > > > > > > Dear reviewer #Dyvx,
> > > > > > >
> > > > > > > Thanks for your great efforts in reviewing our paper.  Your constructive comments have greatly helped us improve our paper.  Do you have any other concerns right now? If you have no further questions/concerns, would you mind raising the score?
> > > > > > >
> > > > > > > Best regards and thanks,
> > > > > > >
> > > > > > > Authors of #1826

---

> > > > > > > ### Author Response · Authors · 2023-11-22
> > > > > > > **Window for discussion and revision is closing**
> > > > > > >
> > > > > > > Dear reviewer #Dyvx,
> > > > > > >
> > > > > > > Do you still have concerns? Please let me know as the window is closing soon.
> > > > > > >
> > > > > > > Best regards and thanks,
> > > > > > >
> > > > > > > Authors of #1826

---

> > > > > > > > ### Comment · Reviewer_Dyvx · 2023-11-23
> > > > > > > > **Q1 has not been addressed**
> > > > > > > >
> > > > > > > > Dear Authors
> > > > > > > >
> > > > > > > > Q1 has not been addressed fully.
> > > > > > > >
> > > > > > > > Q3 is unfortunate, but this is an issue that I did not address in the first review, so I would not base my decision on it. But if this experiment had been successful, this would have been a main plus. The current result is unsatisfactory as the performance on blip is not good in the absolute sense.
> > > > > > > >
> > > > > > > > My remaining concern is only on Q1. But I will incrase the score. I expect the authors to make in detail their experimental findings. I believe showing the limitation is also a contribution.

---

> ### Author Response · Authors · 2023-11-21
>
> Thank you for your prompt responses! Please find our responses below.
>
> >**Q1:** By 'fine-tuning' I mean fine-tuning of the learnable input prompt weights (i.e., 'prompt learning' by LSN). I assume that the set $S$ in the subscript of the expectation in Eqs. (4) and (5) are your training set for prompt learning. As far as I understand, based on your responses comments, this set is not the full ImageNet-1k training set. If you increase the size of $S$ to the full train fold of IN-1k, then would it correspondingly increase the performance, and achieve the SOTA? (By the way, the notations are quite confusing. $S$ is elsewhere defined as the score function while in the Eqns (4) and (5) it seems not the case.)
>
> **A3:** Thank you for the heads up, we have fixed this error in the revised version. We are afraid that even using the full ImageNet-1k dataset we could not achieve the SOTA you mentioned. We think the main reason is that there are too few parameters that can be learnt in prompt learning compared to fine-tuning networks, and further increasing the number of training samples will not lead to performance improvement. We don't use the full ImageNet-1k dataset to learn the prompt due to the fact that it takes too long even if only the prompts are tuned. We estimate that the program takes 15 hours to run one epoch (batchsize is set to 8), which is intolerable. We provide the effect of different taining sample sizes on the OOD detection performance on ImageNet-100 dataset in the Appendix. You can refer to that. It can be seen that when the sample size is large, the performance gain from further increasing the sample size is not significant.
>
> >**Q3:** Hopefully, the LSN works on BLIP2 as well; I am expecting the result.
>
> **A3:** We apologize for the problems we have when we are about to validate LSN on BLIP2. The model structure of BLIP2 consists of three types: blip2, blip2_t5 and blip2_opt (Please see details here: [BLIP2](https://github.com/salesforce/LAVIS/tree/main/projects/blip2)). When we use the smallest model blip2, we find that the model still can not understand the meaning of "not". Only when using the larger model blip2_t5 the model is able to understand the meaning of "not". However, blip2_t5 is only used in image-to-text generation task and caption generation task. We fail to find a suitable prompt that allows BLIP2 on these tasks to generate a suitable score for testing the model's OOD detection ability.  Therefore, we are still only able to use smaller model to perform image-text matching task to get cosine similarity between images and text as the OOD scoring function. In this case, we need to learn negative prompts through prompts learning. To this end, we test the performance of LSN on ImageNet-100 with BLIP. The result is as follows:
>
> |              | iNaturalist |       | SUN   |       | Places |       | Textures |       | Average |       |
> |--------------|-------------|-------|-------|-------|--------|-------|----------|-------|---------|-------|
> | Method       | FPR95       | AUROC | FPR95 | AUROC | FPR95  | AUROC | FPR95    | AUROC | FPR95   | AUROC |
> | MCM       | 46.45       | 93.37 | 78.48 | 81.76 | 77.14  | 81.05 | 48.10    |  90.92 | 62.54   | 86.77 |
> | CoOp    |   27.83     | 95.26 | 45.85 | 88.26 |  47.38 | 87.42 |   36.28  | 92.93 |  39.33  | 90.96 |
> | CoOp + LSN |    16.33   | 96.73 | 39.48 | 90.94 |  41.76 | 90.42 |  31.59   | 94.27 |   32.29 | 93.09 |

---

> ### Author Response · Authors · 2023-11-23
> **Thanks for your swift reply and raising the score**
>
> Dear reviewer #Dyvx,
>
> Thanks for your swift reply despite such a busy period. We sincerely appreciate that you can raise the score. For the experiments on the BLIP, we believe that the results above are not the best results as we don't explore much due to time constraints. We will do more experiments on BLIP and present the results in the paper. As for Q1, it may be a fact at this stage that when the ID dataset is ImageNet-1k, the OOD detection performance of prompt learning on CLIP may not be as good as the performance of ImageNet-1k pre-trained models. For now, this is indeed a shortcoming of such methods based on CLIP models. We will make this clear in our paper.
>
> Best regards and thanks,
>
> Authors of #1826

---

### Meta-Review · Area_Chair_Fpry · 2023-12-13

**Metareview:**

This paper explores Out-of-Distribution (OOD) detection by learning a set of negative prompts for each class. The learned positive prompt (for all classes) and negative prompts (for each class) are leveraged to measure the similarity and dissimilarity in the feature space simultaneously.

Strengths:

The paper is well-written and easy to follow. The proposed methods are well-motivated and have reasonably good empirical performance. During the rebuttal phase, the authors have well-addressed most of the reviewers' concerns.

Weakness:

The novelty of this work is limited compared to existing work. As pointed out by some reviewers, the proposed method can not be well generalized to a full training dataset and BLIP2 models to achieve state-of-the-art performance. We strongly encourage authors to include those limitations in the final version.

**Justification For Why Not Higher Score:**

Based on the limitations of this work, such as limited novelty and inability to generalize to full training datasets and BLIP2 models, the impact of this work is not transformative.

**Justification For Why Not Lower Score:**

This work has sufficient contributions, such as a well-motivated methodology and reasonably good empirical performance.

---

### Decision · Program_Chairs · 2024-01-16

Accept (poster)